# Structures of the mumps virus polymerase complex via cryo-electron microscopy

Tianhao Li[1,2,3,4,8], Mingdong Liu[1,2,3,8], Zhanxi Gu[5,6], Xin Su [1,2,3,7], Yunhui Liu[1,2,3], Jinzhong Lin [7], Yu Zhang [6] & Qing-Tao Shen [1,2,3,4] ✉

The viral polymerase complex, comprising the large protein (L) and phosphoprotein (P), is crucial for both genome replication and transcription in non-segmented negative-strand RNA viruses (nsNSVs), while structures corresponding to these activities remain obscure. Here, we resolved two L–P complex conformations from the mumps virus (MuV), a typical member of nsNSVs, via cryogenic-electron microscopy. One conformation presents all five domains of L forming a continuous RNA tunnel to the methyltransferase domain (MTase), preferably as a transcription state. The other conformation has the appendage averaged out, which is inaccessible to MTase. In both conformations, parallel P tetramers are revealed around MuV L, which, together with structures of other nsNSVs, demonstrates the diverse origins of the L-binding X domain of P. Our study links varying structures of nsNSV polymerase complexes with genome replication and transcription and points to a sliding model for polymerase complexes to advance along the RNA templates.

The non-segmented negative-strand RNA viruses (nsNSVs) contain many pathogens, including the Ebola virus (EBOV), rabies virus (RABV), human respiratory syncytial virus (HRSV), and mumps virus (MuV), which cause severe human disease and even death[1,2]. During the whole viral life cycle, viral genomes are always enwrapped by viral nucleoproteins (N), forming the helical nucleocapsids (NC) for genome protection and encapsulation[3–5]. After viral entry into host cells, another two viral proteins, including large proteins (L) and phosphoproteins (P), along with NC, are released into cytosols[6]. L and P function as the RNA polymerase complex, responsible for replicating and transcribing the viral genome[7–12].

To catalyze the RNA synthesis in both replication and transcription, L sequentially consists of five domains: the RNA-dependent RNA polymerase domain (RdRp), polyribonucleotidyl transferase domain (PRNTase), connector domain (CD), methyltransferase domain (MTase), and C-terminal domain (CTD)[13–15]. As the core module of L,

RdRp and PRNTase take charge of the RNA synthesis and capping[8,16,17]. MTase has methylation activity and is only required for transcription[11,18–20]. RdRp-PRNTase is quite conserved in structure amongst nsNSVs, while CD-MTase-CTD resembles an appendage of RdRp-PRNTase with great structural diversity[15,21–26]. Specifically, CD-MTase-CTD from HRSV, human metapneumovirus (HMPV), and EBOV are not resolved in structures due to the inherent flexibilities[21,22,24,26], and show distinct spatial organizations in vesicular stomatitis Indiana virus (VSIV), RABV, and parainfluenza virus 5 (PIV-5)[15,23,25,27]. Limited by the number of complete L structures and the lack of functional analyses, the relationship between these conformations and RNA synthesis remains elusive.

P is the polymerase cofactor of L for RNA synthesis[28–33]. P harbors an oligomerization domain ($P_{OD}$) for self-oligomerization, and the oligomeric P attaches to RdRp of L, tethering the polymerase to NC to extract the RNA strand for both replication and transcription[34–38].

[1]School of Life Sciences, Department of Chemical Biology, Southern University of Science and Technology, Shenzhen 518055, China. [2]Laboratory for Marine Biology and Biotechnology, Qingdao National Laboratory for Marine Science and Technology, Qingdao 266237, China. [3]Institute for Biological Electron Microscopy, Southern University of Science and Technology, Shenzhen 518055, China. [4]School of Life Science and Technology, ShanghaiTech University, Shanghai 201210, China. [5]University of Chinese Academy of Sciences, Beijing 100049, China. [6]Key Laboratory of Synthetic Biology, Center for Excellence in Molecular Plant Sciences, Institute of Plant Physiology and Ecology, Chinese Academy of Sciences, Shanghai 200032, China. [7]State Key Laboratory of Genetic Engineering, School of Life Sciences, Zhongshan Hospital, Fudan University, Shanghai 200438, China. [8]These authors contributed equally: Tianhao Li, Mingdong Liu. ✉e-mail: shenqt@sustech.edu.cn

All resolved L−P complexes in HRSV, PIV-5, and EBOV reveal four parallel P molecules[21–23,26]. Immediately after $P_{OD}$, there is an X domain ($P_{XD}$) within the C-terminal domain ($P_{CTD}$), which can bind both RdRp and N[23,39]. Remarkably, the $P_{XD}$-binding sites of N show diversities in nsNSVs, from the RNA-binding domain $N_{CORE}$ to the molecular recognition element (MoRE) motif within the C-terminus of N ($N_{TAIL}$)[36,37,39–45]. Since four $P_{OD}$ assemble into a coiled-coil structure, a model was proposed that P tetramer cartwheels on NC during the advance of the polymerase[46–49]. Once the $P_{OD}$ rotates, the L-anchored $P_{XD}$ is assumed to dissociate from RdRp, and $P_{XD}$ from another P will rebind L. The cartwheeling model requires the binding capability of all four $P_{XD}$ for the iterative cycles. However, recent studies demonstrated that P tetramers with one to three impaired $P_{XD}$ still maintain a comparable or even higher bioactivity of RNA synthesis[39]. Even surprisingly, only one N binding-competent $P_{XD}$ in the tetramer is enough for the minigenome transcription[39,50]. All these reach another sliding model that does not require the oligomeric P to undergo rotation. Unfortunately, both models still lack structural evidence, leaving the L−P advance on NC obscure.

The mumps virus, which belongs to the family *Paramyxoviridae*, is a typical member of nsNSVs that causes acute upper respiratory symptoms and parotitis. Despite available vaccines, several regional outbreaks have still occurred worldwide in the past decades[51–53]. Previous studies on MuV N and P indicated some unique mechanisms that are inconsistent with other well-studied species[37,40,54]. $P_{CTD}$ was identified as the sole L-binding region in MuV, while $P_{CTD}$ alone could not form a stable complex with L in the absence of $P_{OD}$[41]. More surprisingly, the recombinant MuV $P_{OD}$ prefers the formation of anti-parallel tetramers (parallel dimers in anti-parallel configuration)[36,41]. These unusual findings on MuV P and L need further verification on the L−P complex, which will enrich the structure pool of polymerase complexes and benefit the comprehensive understanding of the molecular mechanism for replication and transcription.

Here, we resolved two MuV L−P complex conformations via cryogenic-electron microscopy (cryo-EM). One conformation presents all five domains of L, among which CD-MTase-CTD adopts a spatial organization distinct from PIV-5, with a continuous RNA tunnel from RdRp-PRNTase to CD-MTase-CTD, preferably as a transcription state. The other conformation has CD-MTase-CTD averaged out due to the structural flexibility, with its RNA tunnel inaccessible to MTase, unfavorable for genome transcription. Moreover, parallel P tetramers are revealed in MuV L−P complexes, and our atomic model of MuV P helps in building uncertain residues to the C-terminal regions at the front of the X domain of P ($P_{XD}$) in PIV-5, which, together with other structures, demonstrates the diverse origins of L-binding $P_{XD}$ from the P tetramer in nsNSVs.

## Results

### Two conformations of MuV L−P complex
We co-expressed MuV L and P in Sf9 cells. The recombinant proteins were purified by tandem Strep-Tactin affinity, ion exchange, and size-exclusion chromatography (Supplementary Fig. 1a). SDS-PAGE and western blot analyses verified the assembly of MuV L−P complex from full-length individuals (Supplementary Fig. 1b,c), and the de novo RNA synthesis assay further showed the catalytic activity of MuV L−P as the RNA-dependent RNA polymerase (Supplementary Fig. 1d,e). To unveil the architecture of MuV L−P complex, purified proteins were subjected to cryo-EM analyses. Three-dimensional (3D) classification and refinements revealed two distinct conformations: one resembles the density map of VSIV L, with both the body and the appendage visible (termed $L_{integral}$−P); the other only has the body of L (termed $L_{body}$−P), similar to HRSV, HMPV and EBOV[21,22,24,26] (Supplementary Fig. 2). The missing of appendage in $L_{body}$−P indicates the structural flexibility in MuV L−P, as in other nsNSVs.

In MuV $L_{integral}$−P, the appendage is also less resolved compared with the body. To improve the resolution, "annealing," as a facile and robust approach to synchronize proteins[55], was applied to the same batch of purified MuV L−P samples before the grid preparation. Intriguingly, the particle proportion of $L_{integral}$−P increases from 30.9% to 37.5% after annealing (Supplementary Fig. 2). We then combined $L_{integral}$−P particles from both unannealed and annealed particles and finally obtained a 3.02 Å cryo-EM structure (Table 1 and Supplementary Fig. 3a–d). All five domains of L and regions of $P_{OD}$, $P_{Linker2}$, and $P_{CTD}$ involved in L−P interfaces were clearly resolved (Fig. 1a–c and Supplementary Fig. 4a–c). Following the same strategy, $L_{body}$−P was determined at the resolution of 3.01 Å, and only RdRp and PRNTase are visible in L (Fig. 1d,e, Table 1, and Supplementary Fig. 3e,f). The overall architecture of RdRp and PRNTase in $L_{body}$−P is very similar to the counterparts in MuV $L_{integral}$−P, PIV-5, and VSIV[15,23].

Critical for RNA synthesis, many motifs, including GDN ($L_{778–780}$) motif within RdRp and histidine-arginine (HR, $L_{1298–1299}$) motif within PRNTase, are highly conserved in structures among MuV, PIV-5, and VSIV L−P complexes (Fig. 2a). Two flexible loops termed the priming loop and the intrusion loop in MuV PRNTase have the similar orientations with those of PIV-5[23]. Specifically, the intrusion loop projects into the RNA cavity and the priming loop is oriented to the inner wall of MuV PRNTase (Supplementary Fig. 5a). The up-and-down flipping of these two loops is essential for initiating of RNA synthesis[56].

### $L_{integral}$−P as a favorable transcription state
Different from the conserved RdRp-PRNTase, CD-MTase-CTD of MuV $L_{integral}$−P adopts a spatial organization distinct from those of PIV-5, though their individual structures are pretty similar (Fig. 2a, b). The detailed alignment showed that MTase and CTD in MuV $L_{integral}$−P appears as an integral on the PRNTase side instead of the RdRp side compared with PIV-5 L. The overall spatial organization of CD-MTase-CTD in MuV $L_{integral}$−P is surprisingly similar to VSIV L (Fig. 2b). VSIV CD-MTase-CTD is highly flexible unless unphosphorylated P locks their configuration[27,57], while in MuV, no P fragment is resolved around the appendage (Fig. 2b). In MuV $L_{integral}$−P, an α-helix hinge termed Hinge-1 ($L_{1416–1431}$) connecting PRNTase and CD has been visualized (Fig. 1c). It is supposed to provide flexibility to CD positioning via loops flanking it.

Compared with PIV-5, the spatial organization of MuV CD-MTase-CTD renders the helices α53 ($L_{1439–1458}$) and α57 ($L_{1535–1544}$) of CD rotating upward and leaves more space for RNA to access MTase (Supplementary Fig. 6a). Actually, MuV $L_{integral}$−P forms a continuous positively-charged tunnel from the GDN to the K-D-K-E motifs, ideal for the RNA synthesis followed by 5' capping and methylation, which is favorable as the transcription state (Fig. 2c and Supplementary Fig. 6b). While in PIV-5 L, the RNA tunnel towards the K-D-K-E motif is blocked at the site surrounded by RdRp and CD due to the different spatial organization of CD-MTase-CTD (Supplementary Fig. 6c). Furthermore, the K-D-K-E motif of PIV-5 locates at the outside of the RNA cavity. The capped mRNA is hard and even impossible to get access to the methylation site as the transcription state. MuV $L_{body}$−P owns a flexible appendage, and its RNA tunnel is inaccessible to MTase-CTD, which is unfavorable for transcription. However, the RNA cavity formed by RdRp and PRNTase domains is available, with the potential capability for genome replication (Supplementary Fig. 6d).

### Parallel P tetramers in MuV L−P complex
Our $L_{integral}$−P structure revealed four P molecules assembled into a helical bundle around RdRp of L via their respective $P_{OD}$. Different from the previous analysis on MuV P alone[36,41], four P molecules are more likely to adopt a parallel orientation in $L_{integral}$−P (Fig. 3a and Supplementary Fig. 4b,c). This observation is highly consistent with P molecules in many other nsNSVs[21–24,26,34,58–60], which indicates a

**Table 1 | Cryo-EM data collection, refinement, and validation statistics of MuV $L_{integral}$–P and $L_{body}$–P**

| | $L_{integral}$–P | RdRp-PRNTase of $L_{integral}$–P | CD-MTase-CTD of $L_{integral}$–P | P of $L_{integral}$–P | $L_{body}$–P | P of $L_{body}$–P |
|---|---|---|---|---|---|---|
| **Data collection and processing** | | | | | | |
| Voltage (kV) | 300 | 300 | 300 | 300 | 300 | 300 |
| Electron exposure (e$^-$/Å$^2$) | 50 | 50 | 50 | 50 | 50 | 50 |
| Defocus range (µm) | –1.5 to –2.5 | –1.5 to –2.5 | –1.5 to –2.5 | –1.5 to –2.5 | –1.5 to –2.5 | –1.5 to –2.5 |
| Symmetry imposed | C1 | C1 | C1 | C1 | C1 | C1 |
| Initial particle images(no.) | 2,087,570 | 2,087,570 | 2,087,570 | 2,087,570 | 2,087,570 | 2,087,570 |
| Final particle images (no.) | 438,014 | 438,014 | 438,014 | 88,107 | 477,568 | 41,168 |
| Pixel size (Å) | 0.53 | 0.53 | 0.53 | 1.06 | 0.53 | 1.06 |
| Map resolution (Å) | 3.02 | 2.93 | 3.13 | 3.49 | 3.01 | 3.63 |
| FSC threshold | 0.143 | 0.143 | 0.143 | 0.143 | 0.143 | 0.143 |
| Map sharpening B factor (Å$^2$) | 136.8 | 133.6 | 156.1 | 132.3 | 143.6 | 123.1 |
| EMDB code | 37957 | 37959 | 37958 | 37960 | 37961 | 37962 |
| | | Composite map EMD-35864 | | | Composite map EMD-37964 | |
| **Model building and refinement** | | | | | | |
| Initial model used (PDB code) | / | 6V85 | / | 4EIJ | 6V85 | 4EIJ |
| Model composition | | | | | | |
| Non-hydrogen atoms | / | 11,632 | 5,588 | 2,096 | 10,784 | 1,958 |
| Protein residues | / | 1,464 | 699 | 278 | 1,350 | 258 |
| Ligands | / | 2 | 0 | 0 | 2 | 0 |
| B factors (Å$^2$) | | | | | | |
| Protein | / | 61.76 | 76.74 | 115.85 | 58.08 | 130.10 |
| Ligand | / | 147.46 | / | / | 142.51 | / |
| R.m.s deviations | | | | | | |
| Bond lengths (Å) | / | 0.005 | 0.005 | 0.008 | 0.004 | 0.007 |
| Bond angles (°) | / | 0.757 | 0.777 | 1.234 | 0.605 | 1.108 |
| Validation | | | | | | |
| MolProbity score | / | 1.73 | 1.66 | 1.91 | 1.42 | 1.80 |
| Clashscore | / | 10.29 | 14.27 | 17.23 | 7.74 | 16.46 |
| Poor rotamer (%) | / | 1.22 | 0.64 | 1.67 | 0.33 | 0.45 |
| Ramachandran plot | | | | | | |
| Favored (%) | / | 97.31 | 98.09 | 98.52 | 98.21 | 97.60 |
| Allowed (%) | / | 2.69 | 1.91 | 1.48 | 1.79 | 2.40 |
| Disallowed (%) | / | 0.00 | 0.00 | 0.00 | 0.00 | 0.00 |
| PDB code | / | 8YXM | 8YXL | 8YXO | 8YXP | 8YXR |
| | | Composite model PDB ID 8IZL | | | Composite model PDB ID 8X01 | |

generally conserved mechanism for P molecules to mediate RNA genome replication and transcription.

In cryo-EM maps, four P molecules assemble like a kettle spout stably anchored to L. Our cryo-EM structures capture the clear interface between two P molecules (depicted as P1 and P4, respectively) and L, involving the RdRp, $P_{OD}$, $P_{Linker2}$, and $P_{CTD}$ regions (Fig. 3a). Specifically, P1-Met$_{269}$, P1-Val$_{273}$, P4-Ala$_{271}$, and P4-Val$_{273}$ from the C-terminus of the $P_{OD}$ core region, form a hydrophobic cap to trap the conserved residue L-Phe$_{394}$ (Fig. 3b). A salt bridge between P4-Glu$_{267}$ and L-Lys$_{453}$, together with a hydrogen bond between P1-Thr$_{265}$ and L-Asn$_{428}$, further stabilizes this interface (Fig. 3b). The $P_{OD}$ tail of P4 folds into a β-strand, forming the anti-parallel β-sheet with Lys$_{390}$–Asp$_{393}$ of L (Fig. 3c). Three electrostatic interactions fix both ends of the β-sheet. Furthermore, L-Gln$_{680}$ forms hydrogen bonds with P4-Met$_{276}$ and P4-Asp$_{277}$, enabling close contact between these two β-strands (Fig. 3c).

Intriguingly, the C-terminal domains of P1 and P4 turn to the template entry side (Fig. 3a). The turning point occurs at the $P_{Linker2}$ region of P4 and is trapped in the hydrophobic groove contributed by L-Ala$_{731}$, L-Leu$_{732}$, and L-Val$_{739}$. Hydrogen bonds and salt bridges stabilize residues flanking the turning point (Fig. 3d). Several hydrogen bonds make an interacting network among the $P_{Linker2}$ of P1, $P_{CTD}$ of P4, and RdRp of L (Fig. 3e). The $P_{CTD}$ of P4 strides over P1, forming hydrogen bonds among P4-Ser$_{301}$, L-Arg$_{459}$, and L-Arg$_{687}$. The $P_{Linker2}$ of P1 is surrounded by both L and P4. P1-Thr$_{282}$ and P1-Val$_{284}$ contact with RdRp, and P1-Pro$_{281}$ and P1-Gly$_{283}$ interact with the CTD of P4 (Fig. 3e).

Apparently, L–P binding involves abundant residues via forming a complicated and stable interface[39,61]. Residues from 249 to 299 of P are the major region interacting with L (Supplementary Fig. 7). Interestingly, these fragments in six different MuV strains are identical while other regions are not (Fig. 3f and Supplementary Fig. 8). This indicates that these residues are evolutionarily highly conserved, and play critical roles in the stable assembly of the L–P complex.

## Diverse origins of L-binding $P_{XD}$

RNA synthesis requires the advance of L on NC, both of which are tethered by different $P_{XD}$ in the P tetramer[39]. In PIV-5, one $P_{XD}$ binds to

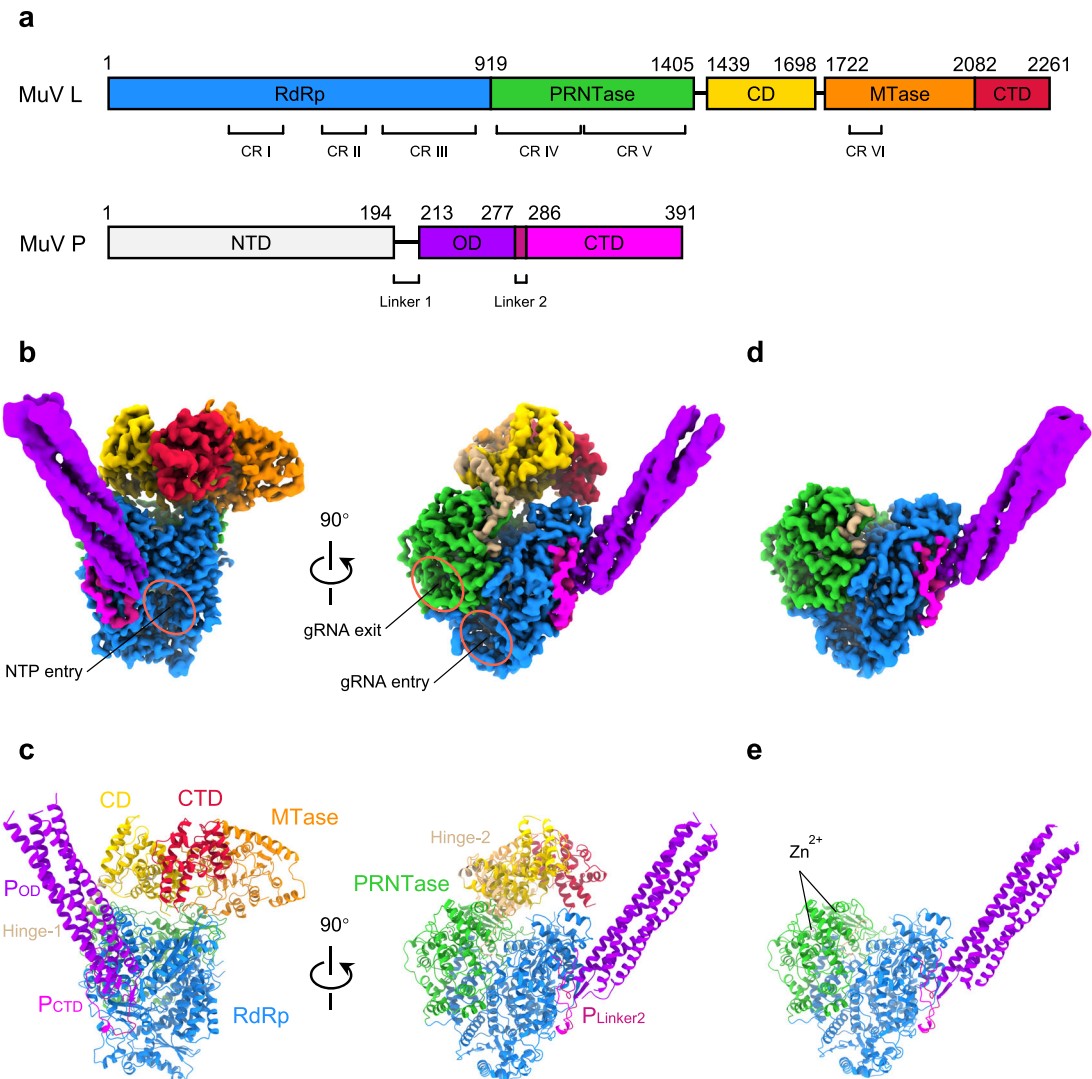

**Fig. 1 | Structures of MuV L–P complex. a** Diagram of MuV L and P domains. RdRp, PRNTase, Hinge regions (Hinge-1&2), CD, MTase, and CTD of L are colored in blue, green, tan, gold, orange, and crimson, respectively. NTD, OD, Linker region 2 (Linker2), and CTD of P are colored in light gray, purple, violet red, and magenta, respectively. The same color strategy is used throughout the manuscript unless specified. CR I–VI: six conserved regions in L. **b** Cryo-EM density maps of MuV $L_{integral}$–P (EMD-35864). NTP entry, genomic RNA (gRNA) entry, and gRNA exit are circled. **c** Atomic models of MuV $L_{integral}$–P (PDB ID 8IZL). **d** Cryo-EM density map of MuV $L_{body}$–P (EMD-37964). **e** Atomic model of MuV $L_{body}$–P (PDB ID 8X01). Maps in (**b**, **d**) are the composite cryo-EM maps of MuV $L_{integral}$–P and $L_{body}$–P to improve the interpretability, after post-processing in DeepEMhancer[69]. These are also utilized for other figure preparation. Models in (**c**, **e**) are the composite atomic models of MuV $L_{integral}$–P and $L_{body}$–P via rigid body docking of individual models into their respective composite cryo-EM maps, which are B-factor sharpened.

RdRp as the major contact site, preventing the detachment of P from the L–P complex[23]. There are four P molecules in PIV-5 L–P, and the exact origin of this L-binding $P_{XD}$ remains vague due to the poor densities of $P_{OD}$ tail, $P_{Linker2}$, and $P_{CTD}$. In reference to the HRSV L–P structure, the authors speculated that this L-binding $P_{XD}$ in PIV-5 belongs to P4 as well[21,23].

PIV-5 has high sequence identity (L: 58.7%; P: 37.0%) and structural similarity with MuV (Supplementary Fig. 9). Thus, we docked the atomic model of MuV P into the EM density of PIV-5 P (EMD-21095). Interestingly, MuV P1 and P4 fit well in the density map of PIV-5 (Supplementary Fig. 4d). Our intensive model building on PIV-5 identifies that C-terminal domains of P1 and P4 orient to the template entry side instead of the NTP entry side (Fig. 4a). In PIV-5, the L-binding $P_{XD}$ on the NTP entry side should not belong to P4, but P2 (Fig. 4a, b). An extensive survey on the origin of $P_{XD}$ among nsNSVs shows that the L-binding $P_{XD}$ of EBOV belongs to P1 (Fig. 4c), while L-binding $P_{XD}$-like

regions of HRSV and HMPV belong to P4[21,22,24,26] (Fig. 4d). Apparently, L-binding $P_{XD}$ has diverse origins.

The traditional cartwheeling model assumes that the relative position of L-binding $P_{XD}$ and its corresponding $P_{OD}$ stays the same during rotation cycles[38,47–50]. The origin diversity of $P_{XD}$ mentioned above seemingly supports an optimized cartwheeling mechanism, by which the rotation of $P_{OD}$ does not interfere with the stable binding of one single $P_{XD}$ to RdRp (Fig. 5a). Nevertheless, after several rotation cycles, the coiling tension will accumulate in L-binding $P_{XD}$, which may break the stable interface formed by $P_{XD}$ and RdRp and further hamper the binding sustainability.

Different from the cartwheeling model, another popular sliding model claims that any $P_{XD}$ in tetrameric P can stably bind to RdRp, and other $P_{XD}$ will reengage with RdRp only after the falling-off of the current $P_{XD}$ from L[39]. Diverse origins of $P_{XD}$ in nsNSVs are well consistent with the proposed free competition among all four L binding-

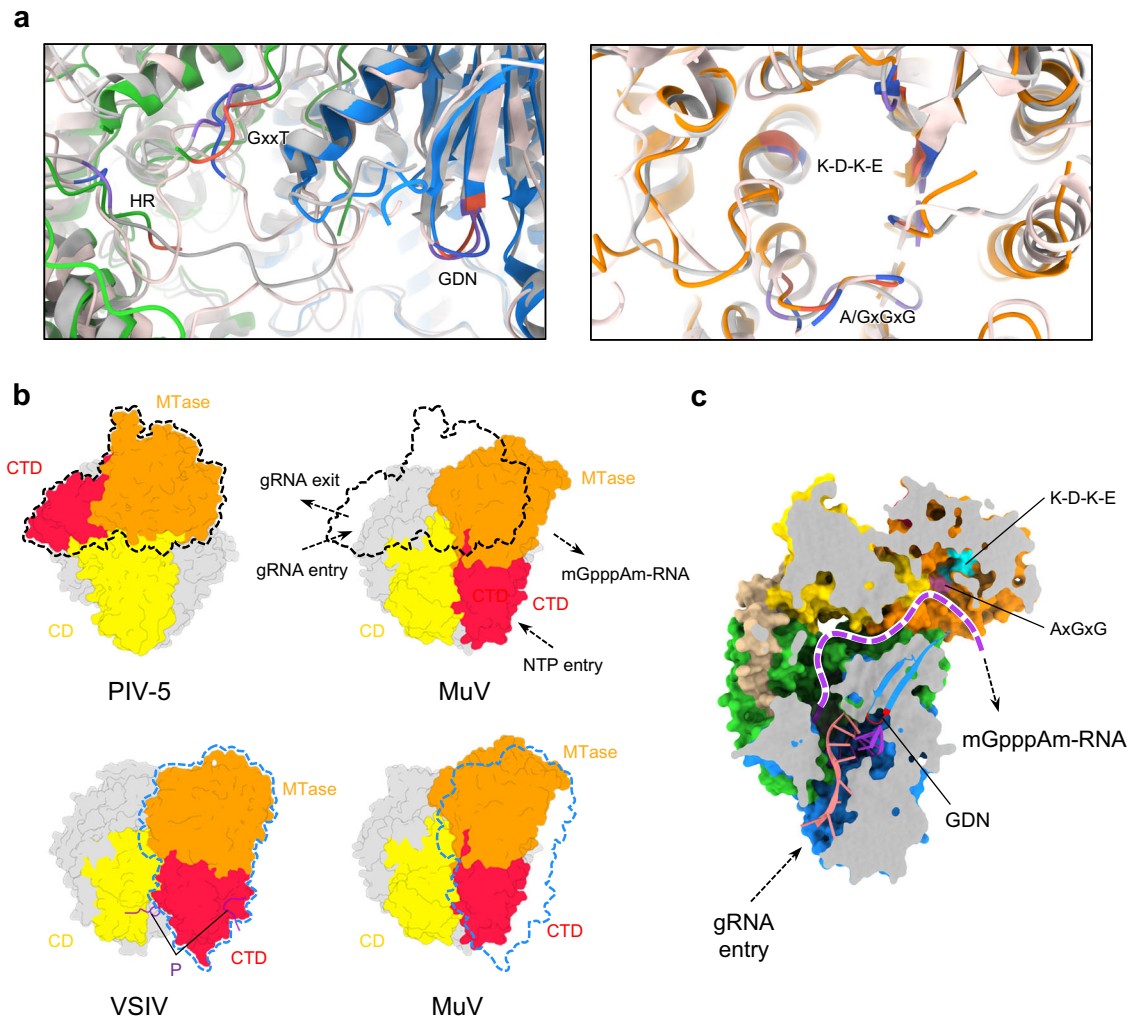

**Fig. 2 | MuV L$_{integral}$–P as a favorable transcription state. a** Comparison of critical motifs among MuV, PIV-5, and VSIV L. Motifs (GDN, GxxT, HR, K-D-K-E, and A/GxGxG) of MuV, PIV-5, and VSIV are colored in tomato, medium purple, and royal blue, respectively; the other parts of PIV-5 and VSIV are colored in silver and misty rose, respectively. **b** Comparison of CD-MTase-CTD spatial organizations among MuV, PIV-5, and VSIV L. RdRp and PRNTase of all three structures are aligned and colored in light gray. The P fragment of VSIV is colored in purple. The outlines of PIV-5 and VSIV MTase-CTD are depicted in black and blue dashed lines around MuV maps, respectively. **c** Continuous RNA tunnel of MuV L$_{integral}$–P. Superposed nucleotides are from the crystal structure of the reovirus λ3 polymerase initiation complex (PDB ID 1N1H). The purple dashed curve represents the potential elongation path for the transcribed mRNA.

competent P$_{XD}$ (Fig. 5b). The other P$_{XD}$ dynamically binds to nucleoproteins, ensuring the processivity of the RNA synthesis.

## Discussion

L–P complex is responsible for RNA synthesis in both replication and transcription processes in nsNSVs. As the core component of the L–P complex, one L structure is usually resolved from each viral species. However, L differs in structure among different species, especially on the spatial organization of CD-MTase-CTD. Via cryo-EM, we resolved two conformations of MuV L–P complex: L$_{body}$–P and L$_{integral}$–P. MuV L$_{integral}$–P adopts a different spatial organization of CD, MTase, and CTD from PIV-5 L–P and possesses a continuous RNA tunnel as the transcriptionally competent form. The proximity of MTase during the elongation of the transcription ensures the regulation of both the methylation and polyadenylation process[20,62].

Once the polymerase adopts the replication state, the K-D-K-E motif and SAM-binding motif in MTase of L are no longer required. MuV L$_{integral}$–P may bypass the MTase domain as one possible replication form (Supplementary Fig. 10a). MuV L$_{body}$–P takes an appendage-free conformation but still owns an exposed RNA cavity formed by RdRp and PRNTase domains with the potential to another

replication state for RNA synthesis (Supplementary Fig. 6d and 10b). In PIV-5, the RNA tunnel to MTase is blocked, while RNA can still come out from the pore formed by RdRp and CD. Thus, the conformation revealed in PIV-5 might be the third form as the replication state (Supplementary Fig. 10c). Validation of different forms for replication or transcription needs the further setup of functional assays on various mutants.

In both replication and transcription processes, the priming loop and the intrusion loop are critical to regulating RNA synthesis[56]. Both MuV and PIV-5 belong to the family *Paramyxoviridae* and have the exact position in the priming loop and intrusion loop. Other species within the same family also share the conserved positions of these two loops (Supplementary Fig. 11). To date, HRSV and HMPV in *Pneumoviridae* harbor both up-flipping loops[21,24] (Supplementary Fig. 5b). The priming loops of VSIV and RABV in the *Rhabdoviridae* occupy the cavity, while the intrusion loops closely attach to PRNTase[15,25] (Supplementary Fig. 5c). We hypothesize that the genetic diversity among families results in textural differences in cavities, further leading to these three preferred organizations. Previous studies proposed the possible connection between the positions of two loops and the polymerase states of RNA synthesis. Based on the EBOV P–L–RNA

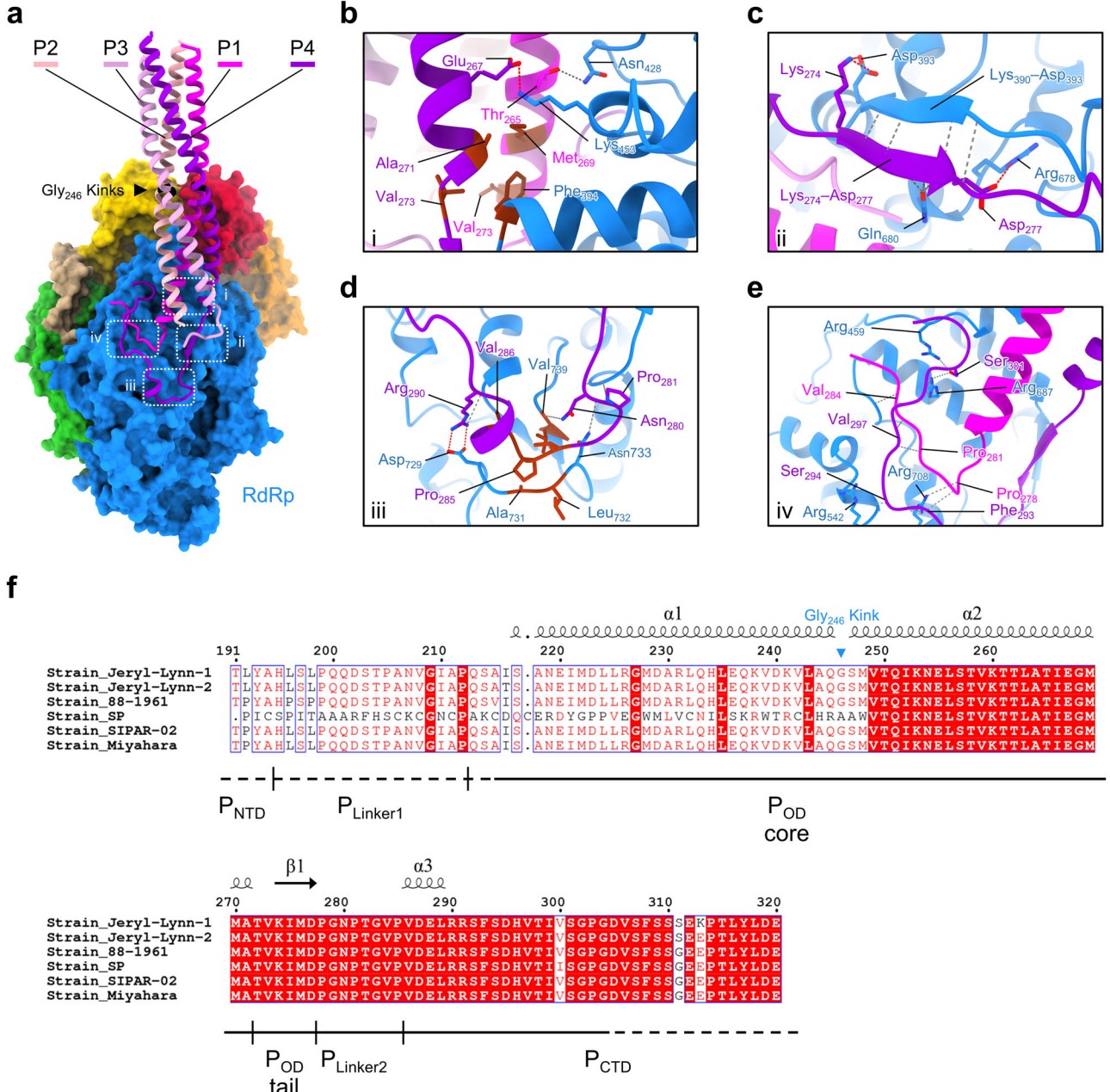

**Fig. 3 | The interface between L an P of MuV polymerase complex. a** Overall view of tetrameric P bound to RdRp of L. P1, P2, P3, and P4 are colored in magenta, light pink, plum, and dark violet, respectively. Four core interaction zones are boxed with white dotted rectangles. **b–e**, Close-up views of the interaction zones i, ii, iii, and iv. Hydrogen bonds and salt bridges are indicated by dim gray and red dashed lines, respectively. Residues involved in hydrophobic interactions are colored in brown. **f** Sequence alignment of P from six MuV strains spanning residues 191–320. Solid lines beneath the sequences represent the structurally resolved regions. Dashed lines represent the unsolved regions. Labels of secondary structures above the sequences are based on the atomic model of P4. $Gly_{246}$, the kink between helices α1 and α2 of $P_{OD}$, is labeled.

structure, during the elongation state, the priming loop and the intrusion loop retract from the cavity to accommodate RNA[26] (Supplementary Fig. 5d). For the resting state, two loops of apo-L either adopt those three stabilized patterns or wobble in the empty cavity, waiting for the incoming NTP and RNA to transform into the pre-initiation state resembling L structures of VSIV and RABV[25,27].

P is required for RNA synthesis in most nsNSVs. MuV P forms parallel dimers and further self-assembles into anti-parallel tetramers in the case of the recombinant $P_{OD}$[36,41]. In this study, we observed that P tetramerizes more probably in a parallel pattern when constituted in complex with L. Due to the moderate resolution of P tetramer, we could not recognize the kink at $Gly_{246}$, the unique feature to identify

the helix orientation, and we could not rule out the possibility of MuV P tetramer in an anti-parallel configuration. The oligomerization forms of P in nsNSVs may depend on different conditions. Nipah virus $P_{OD}$ assembles into trimers in solution but is crystallized into tetramers[34,60,63]. Crystal structures of the Zaire ebolavirus VP35 oligomerization domain are trimers, whereas VP35 forms tetramers in polymerase complexes[26,64]. Therefore, the involvement of L may guide the assembly of P monomers in EBOV, MuV, and others.

Based on parallel P tetramers in nsNSVs L–P complexes, the cartwheeling or sliding model has been proposed to describe the advance of polymerase on NC[39,46–49]. In this study, diverse origins of $P_{XD}$ in nsNSVs provide direct clues to the rotation of P molecules,

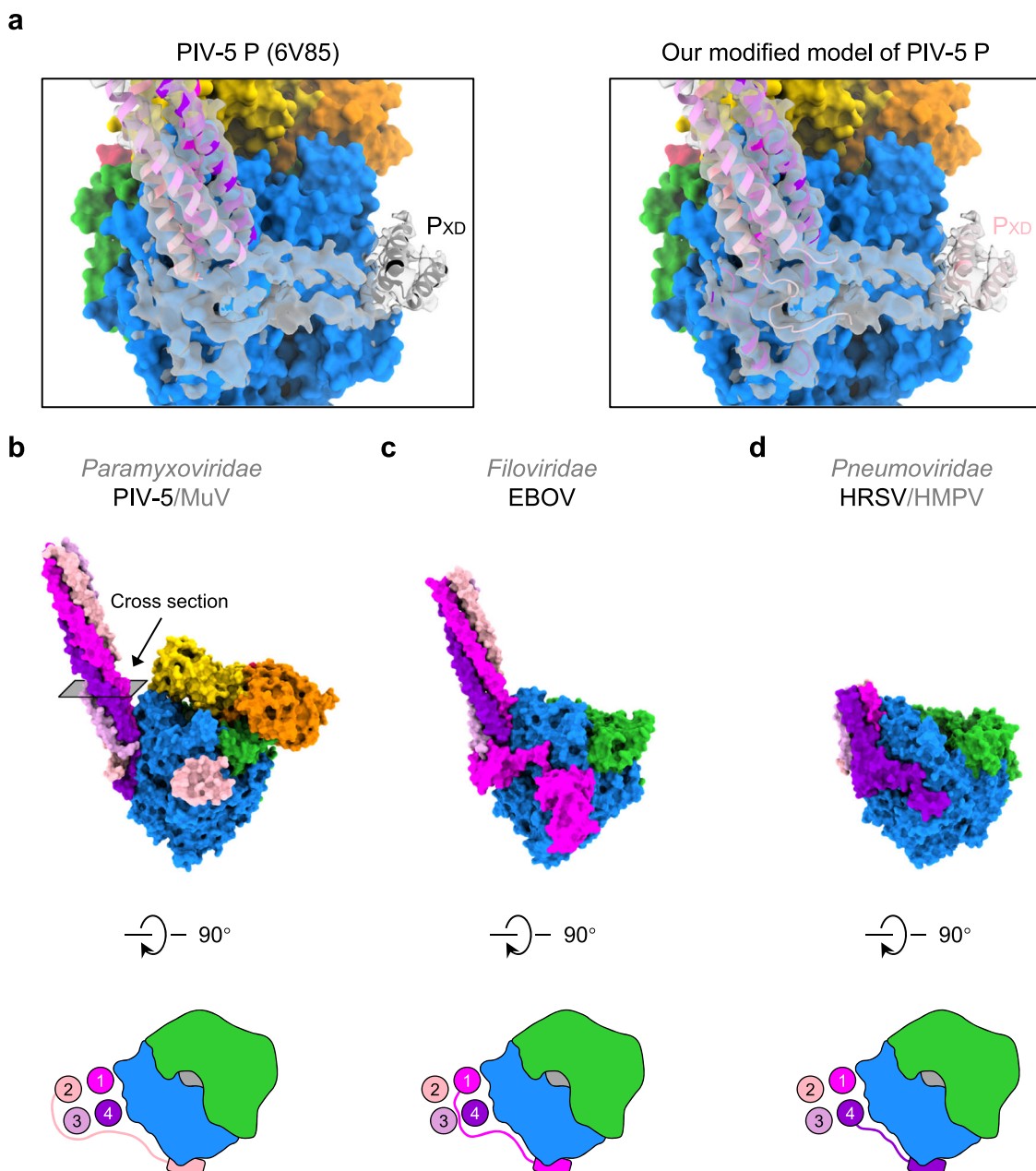

**a**

PIV-5 P (6V85)                    Our modified model of PIV-5 P

$P_{XD}$                              $P_{XD}$

**b** *Paramyxoviridae*    **c** *Filoviridae*    **d** *Pneumoviridae*
PIV-5/MuV                 EBOV                HRSV/HMPV

Cross section

90°                          90°                          90°

**Fig. 4 | Diverse origins of L-binding $P_{XD}$ in nsNSVs. a** The atomic model of PIV-5 P (PDB ID 6V85) docked into the PIV-5 P density (EMD-21095) (Left) and the modified atomic model of PIV-5 P based on the atomic model of MuV P (Right). **b** The side view of our newly built atomic model of PIV-5 L–P complex. The cartoon demonstrates the top view of the L–P interface. Four circles represent the cross-sections of the P tetramer. The rounded rectangle represents the $P_{XD}$. **c** The side view of the atomic model of the EBOV L–P complex. The top view of the L–P interface is shown in the cartoon style. **d** The side view of the atomic model of the HRSV L–P complex. The top view of the L–P interface is shown in the cartoon style.

though from different species. Compared with the cartwheeling model, the sliding model seems more plausible based on current biochemical and structural evidence. The comprehensive structural analyses on the L–P–N–RNA super-complex with modified L or P will be helpful in verifying either the cartwheeling or sliding model. Further studies will shed light on anti-viral drug discovery and eventually benefit human health.

## Methods

### Protein expression and purification
The mumps virus (strain Jeryl-Lynn) L gene (Genbank: AAF70396.1) with an N-terminal dual Strep-tag II and/or MuV P gene (Genbank: AAF70389.1) with a C-terminal Flag-tag were subcloned into the

pFastBac Dual vector and expressed in Sf9 cells (Invitrogen, USA). Cells expressing L (for de novo RNA synthesis assay) or the L–P complex were lysed by Dounce homogenization in Lysis Buffer (300 mM NaCl, 50 mM Tris-HCl, 6 mM $MgCl_2$, and 1 mM TCEP, pH 8.0) supplemented with EDTA-free protease inhibitor cocktail (Bimake, USA). After the high-speed centrifugation at $100,000 \times g$ for 40 min, the supernatant was incubated with Strep-Tactin (Cytiva, USA) resins for 30 min at 4 °C. The resins were washed using Lysis Buffer and eluted using Elution Buffer (2.5 mM d-Desthiobiotin, 150 mM NaCl, 20 mM Tris-HCl, 6 mM $MgCl_2$, and 1 mM TCEP, pH 8.0). The eluted L–P complex was further purified using the captoQ ImpRes column (Cytiva, USA). The fractions containing the L–P complex were concentrated and loaded onto the Superose 6 Increase

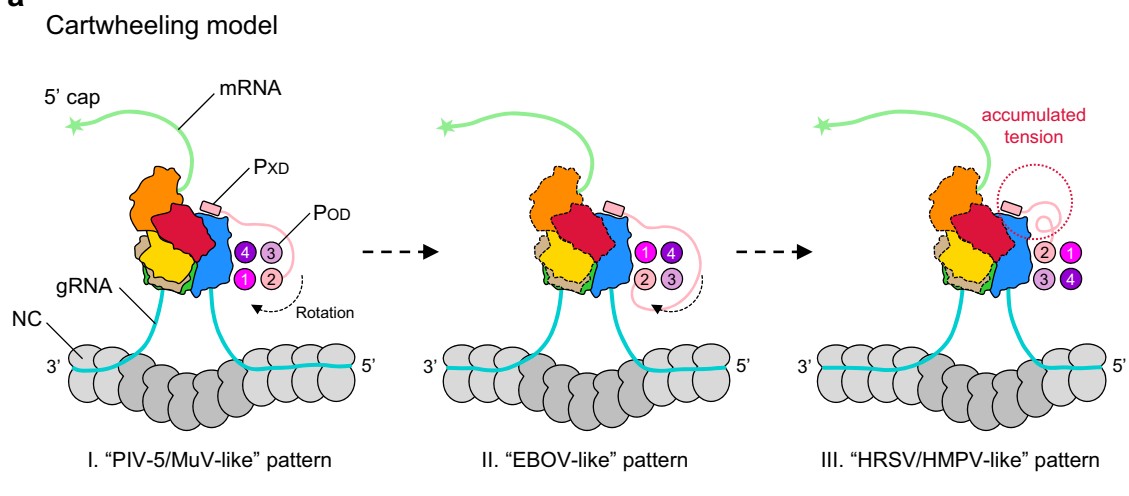

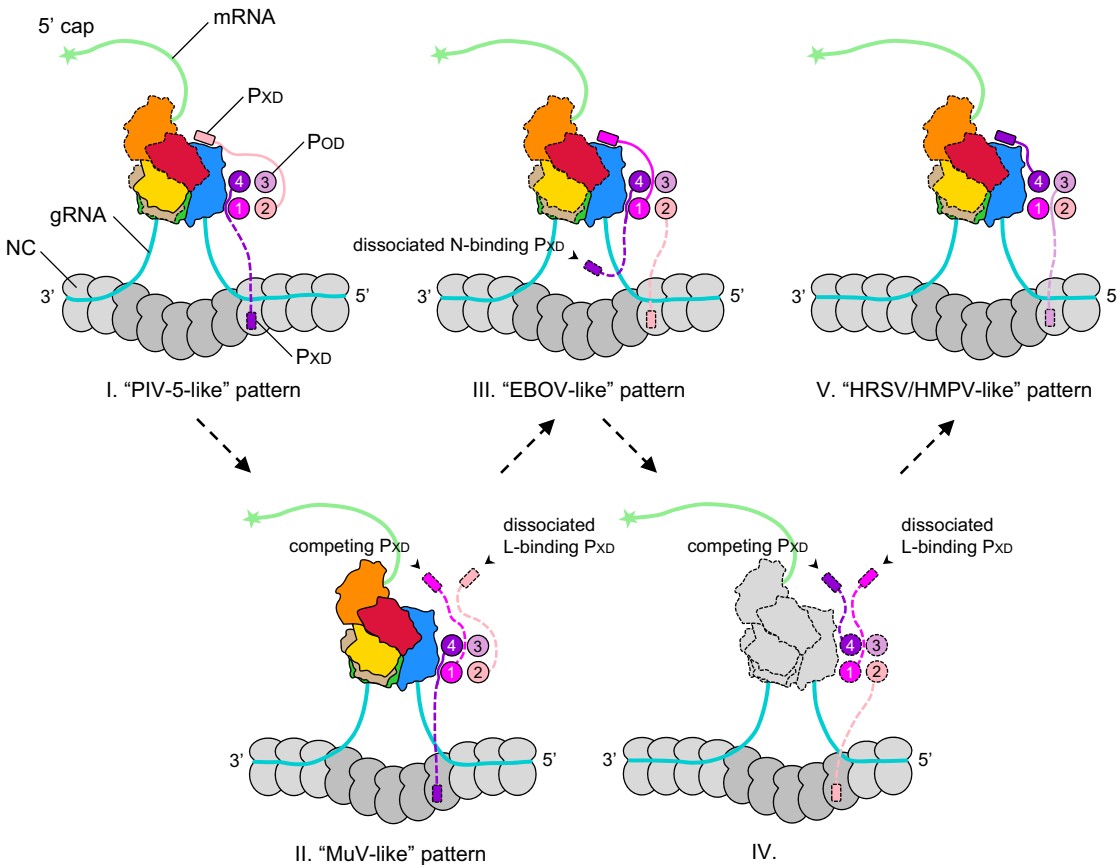

**Fig. 5 | Models for RNA synthesis of nsNSVs. a** The revised cartwheeling model. Tetrameric $P_{OD}$ self-rotates during the advance of L along the RNA template, with the same $P_{XD}$ attached to RdRp. Meanwhile, other $P_{XD}$ may dynamically bind to N to cartwheel on NC. After several rounds, the coiling tension of the $P_{CTD}$ region will arise. The L–P complexes shown here are observed from the top view of the architecture in Fig. 1b. Domains that have not yet been determined are outlined in dashed lines. **b** The sliding model. One $P_{XD}$ binds to RdRp while the other $P_{XD}$ attaches to NC. During the advance of L, the L-binding $P_{XD}$ may be released, and then one of the four $P_{XD}$ competes to rebind L. Once the new L-$P_{XD}$ interaction forms, the remaining $P_{XD}$ may bind N in proximity, followed by cycle repeats.

column (Cytiva, USA) equilibrated in SEC Buffer (150 mM NaCl, 20 mM Tris-HCl, 6 mM $MgCl_2$, and 1 mM TCEP, pH 8.0). L and P in the purified complex were verified by the western blotting using the Strep-Tactin horse radish peroxidase conjugate at the dilution of 1:100,000 (IBA Lifesciences, Germany, catalog number 2-1502-001) and mouse monoclonal antibody against the Flag tag at the dilution of 1:1,000 (Sigma, USA, catalog number F1804), respectively. Samples were concentrated, flash-frozen, and stored at −80 °C.

## De novo RNA synthesis assay

De novo RNA synthesis assays were carried out using 200 nM MuV L or L–P complex and 200 nM templates (Le18, 5′-AUUCAUUCUCCCCUUGGU-3′; Tr18, 5′-ACCAAGGGGAGAAAGUAA-3′) as a reaction mixture in the buffer containing 20 mM Tris·HCl, 100 mM NaCl, 6 mM $MgCl_2$, and 1 mM TCEP, pH 8.0. The reaction mixtures were incubated for 10 min at room temperature. Reactions were initiated through the addition of an NTP mix (final concentrations, 100 µM each of ATP, UTP, and CTP, 1 µM GTP) and 4 µCi [α-$^{32}$P] GTP (3,000 Ci/mmol; Perkin Elmer, USA), allowed to proceed for 3 h at 30 °C, and then stopped by the addition of 5 µL Stop Buffer (8 M urea, 20 mM EDTA, 0.025% xylene cyanol, and 0.025% bromophenol blue). The samples were boiled for 5 min and immediately cooled on ice for another 5 min, followed by running on a 23% (19:1 acrylamide/bisacrylamide) urea polyacrylamide slab gels in 90 mM Tris-borate (pH 8.0) and 0.2 mM EDTA. The radiograph was obtained by storage-phosphor scanning (Typhoon; Cytiva, USA).

## Primer-extension assay

Primer-extension assays were carried out using 200 nM MuV L–P complex and 200 nM template (Le18, 5′-AUUCAUUCUCCCCUUGGU-3′) in a reaction mixture containing 20 mM Tris·HCl, 100 mM NaCl, 6 mM $MgCl_2$, and 1 mM TCEP, pH 8.0. The reaction mixtures were incubated for 10 min at room temperature and then supplemented with the primer (5′-pACCA-3′; final concentration, 1 µM) followed by incubation for 10 min at room temperature. Reactions were initiated by adding 4 µCi [α-$^{32}$P] GTP (3,000 Ci/mmol; Perkin Elmer, USA) and one of the NTP sets: GTP (final concentration, 1 µM GTP), ATP + GTP (final concentrations, 100 µM ATP and 1 µM GTP), UTP + ATP + GTP (final concentrations, 100 µM each of ATP and UTP, and 1 µM GTP). Reactions were allowed to proceed for 3 h at 30 °C and then stopped by adding 5 µL Stop Buffer. The samples were boiled for 5 min and immediately cooled on ice for another 5 min, followed by running on a 23% (19:1 acrylamide/bisacrylamide) urea polyacrylamide slab gels in 90 mM Tris-borate (pH 8.0) and 0.2 mM EDTA. The radiograph was obtained by storage-phosphor scanning (Typhoon; Cytiva, USA).

## Cryo-EM sample preparation

MuV L–P complex at 2.0 µM was melted on ice. Part of the sample was kept as the untreated sample under 4 °C for vitrification. The other part of the sample was subjected to the annealing treatment as described[55]. Specifically, no more than 20 µL of the aliquot was pipetted into one PCR tube and heated in the 37 °C water bath for 1 min. The heated sample was immediately immersed in a mixture of salt, ice, and water (measured temperature: −18 °C) for 20 s and then transferred into the ice bath (measured temperature: 0 °C) for 2 min.

The annealed and unannealed samples were applied to glow-discharged holey grids R2/1 (Quantifoil, Ted Pella, USA). The grids were blotted using a Vitrobot Mark IV (Thermo Fisher Scientific, USA) with 1 s blotting time, force level of 0, and humidity of 100% at 4 °C, and then immediately plunged into liquid ethane and transferred to liquid nitrogen for future cryo-EM imaging.

## Cryo-EM data collection

Data collection was performed with the Titan Krios G$^{3i}$ microscope (Thermo Fisher Scientific, USA) equipped with a K3 BioQuantum direct electron detector (Gatan, USA). Movies were collected via FEI EPU (Thermo Fisher Scientific, USA) automated data collection software at a total dose of ~50 e$^-$/Å$^2$ fractionated over 40 frames with defocus values ranging from −1.5 to −2.5 µm. A super-resolution mode was used with the final pixel size at 0.53 Å.

## Cryo-EM data processing

The raw movie stacks of both the annealed group and the unannealed group were aligned and summed in accordance with dose weighting with MotionCor2.1[65]. The contrast transfer function (CTF) parameters of the summed micrographs were determined with CTFFIND4[66]. Micrographs of two groups with maximum resolution estimates better than 5 Å were imported into CryoSPARC v3.1, respectively[67]. Automatic particle picking was performed on the selected micrographs, and particle sets were created and subjected to reference-free 2D classifications. Obvious junks were excluded from the particle set. After rounds of 2D classifications, 1,172,627 particles (the annealed group) and 914,943 particles (the unannealed group) were selected for the *Ab-Initio* 3D reconstruction, respectively.

Two of three classes for each group were selected as reference structures for the following heterogeneous refinement. In both groups, one class (termed L$_{integral}$–P) contains more structural information about L, while the other class (termed L$_{body}$–P) contains less. The particle proportion of L$_{integral}$–P in the annealed group is 37.5%, while in the unannealed group is 30.9%. The class L$_{integral}$–P in both groups shows no noticeable difference in structure. Therefore, we combined the L$_{integral}$–P or L$_{body}$–P datasets from both groups to improve the resolution. After B-factor sharpening, the respective resolutions of L$_{integral}$–P and L$_{body}$–P were estimated at 3.02 Å and 3.01 Å, based on the gold-standard Fourier shell correlation (FSC) 0.143.

To improve the resolution of L$_{integral}$–P, the local refinements were applied on the body and the appendage via generating focused maps. P tetramers in both L$_{integral}$–P and L$_{body}$–P were poorly resolved; particles were re-extracted using P as the box centers and then subjected to 3D refinements. More cryo-EM densities of P were visible in L$_{integral}$–P and L$_{body}$–P. The locally refined maps, including the body, the appendage, and P tetramers of L$_{integral}$–P, were B-factor sharpened at their respective resolutions of 2.93 Å, 3.13 Å, and 3.49 Å. The locally refined map of P tetramers of L$_{body}$–P was estimated at the resolution of 3.63 Å.

For both L$_{integral}$–P and L$_{body}$–P, we combined their respective globally and locally refined maps, including the body, P tetramers, and/or the appendage, into the composite maps using the phenix.combine_focused_map in PHENIX 1.20.1[68]. These composite maps were B-factor sharpened for the rigid body docking of individual atomic models or post-processed using the DeepEMhancer to improve their interpretability for figure preparation[69]. The DeepEMhancer processed maps, together with their locally refined maps, were deposited in the EMDB.

## Model building and structural analysis

The homology model of MuV RdRp and PRNTase of L was generated using PIV-5 L (PDB ID 6V85) as the reference in SWISS-MODEL[70], and the model of CD-MTase-CTD was predicted by RoseTTAFold[71]. These two models were separately docked as rigid bodies into the locally refined maps of L$_{integral}$–P and L$_{body}$–P using UCSF ChimeraX 1.5[72], manually adjusted in COOT 0.9.7[73], and real-space refined against their respective locally refined maps in PHENIX 1.20.1[68]. The stereochemical quality of each model was assessed using MolProbity[74].

The crystal structure of MuV (strain 88-1961) P$_{OD}$ (PDB ID 4EIJ) was used to guide the manual building of MuV P. The rigid body docking of the anti-parallel tetrameric P$_{OD}$ crystal structure was attempted. P$_{OD}$ core fragments (P$_{249–271}$) of P1 and P4 were successfully docked, but the helices of P2 and P3 failed to fit accurately into the density. We then inverted the orientation of the P2–P3 dimer, yielding a reasonable coordinate of the parallel P tetramer. The final coordinates were real-space refined against their respective locally refined maps in PHENIX 1.20.1[68]. The stereochemical quality of each model was assessed using MolProbity[74]. The atomic models of P tetramer, together with the body and/or the appendage, were docked as rigid bodies into their respective composite maps of L$_{integral}$–P and L$_{body}$–P with the assistance of

their globally refined maps; the composite atomic models for $L_{integral}$−P and $L_{body}$−P were built.

The ambiguous P density of PIV-5 was also built based on our homologous structure of MuV P. The PIV-5 $P_{OD}$ tetramer was shifted towards its N-terminus direction for one turn of the α-helix. Four fragments of varying lengths were extended from the end of the $P_{OD}$ core ($P_{201-272}$) to fit the previously unmodeled density.

Structural analyses, including surface electrostatic distribution and structural superimposition, were fulfilled in UCSF ChimeraX. The L−P interface was analyzed using the PDBePISA 1.48[75].

Protein sequences were aligned by Clustal Omega[76] and presented using ESPript 3.0[77]. The phylogenetic tree was generated via the neighbor-joining method with bootstrap values determined by 1000 replicates in MEGA 11[78].

### Reporting summary

Further information on research design is available in the Nature Portfolio Reporting Summary linked to this article.

## Data availability

The cryo-EM density maps, including the globally refined maps, locally refined maps, and the DeepEMhancer processed composite maps, have been deposited to the Electron Microscopy Data Bank (EMDB, https://www.ebi.ac.uk/pdbe/emdb/). The atomic coordinates corresponding to the locally refined maps and the composite maps of MuV $L_{integral}$−P and $L_{body}$−P have been deposited to the Protein Data Bank (PDB, https://www.rcsb.org/). The accession numbers are listed as follows: EMD-37957 ($L_{integral}$−P as the whole), EMD-37959 and 8YXM (RdRp-PRNTase of $L_{integral}$−P), EMD-37958 and 8YXL (CD-MTase-CTD of $L_{integral}$−P), EMD-37960 and 8YXO (P of $L_{integral}$−P), and EMD-35864 and 8IZL (the composite map of $L_{integral}$−P from EMD-37959, EMD-37958, and EMD-37960 and the composite model from PDB IDs 8YXM, 8YXL and 8YXO); EMD-37961 and 8YXP ($L_{body}$−P as the whole), EMD-37962 and 8YXR (P of $L_{body}$−P), and EMD-37964 and 8X01 (the composite map of $L_{body}$−P from EMD-37961 and EMD-37962 and the composite model from PDB IDs 8YXP and 8YXR). Details are listed in Table 1 and Supplementary Fig. 2. All other data is available in the main manuscript file and/or the supplementary information. Source data are provided with this paper.

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

## Acknowledgements

We thank Lei Qi from the CryoEM facility at Shandong University and Kang Li from the CryoEM facility for Marine Biology at QNLM for our cryo-EM data collection. This work was supported by the National Key R&D Program of China, 2021YFF1200400 (Q.-T.S.) and the National Natural Science Foundation of China, 32241028 (Q.-T.S.). Q.-T.S. is an investigator of SUSTech Institute for Biological Electron Microscopy.

## Author contributions

Q.-T.S. designed and supervised research; T.L. performed protein expression and purification; T.L., M.L., X.S., Y.L., and J.L. did cryo-EM data collection and image processing; Z.G. and Y.Z. conducted the RNA synthesis assay; T.L., M.L., and Q.-T.S. analyzed data; T.L. and Q.-T.S. wrote the paper with input from other authors.

## Competing interests

The authors declare no competing interests.
