## [Peer Review File · Nature Communications]

Structures of the mumps virus polymerase complex via cryo-electron microscopyREVIEWER COMMENTS

Reviewer #1 (Remarks to the Author):

The authors reported the cryoEM structure of the L-P complex of mumps virus. The L-P complex is the viral RdRp responsible for viral RNA synthesis using the nucleocapsid as the template. The structural results are important because mumps virus is a re-emerging human pathogen. There are frequent outbreaks in recent years, up to thousands of cases per year.

The primary finding of this work is that the viral RdRp of mumps virus has a very similar structure as that in other important human pathogens, such as PIV5, RSV, EBOV and RABV. This confirms that antiviral approaches to other nsNSVs are also applicable to MuV.

However, the presentation of their results suffers from overclaiming the significance. Many statements are not based on the actual observations.

1. "Coordination of replication and transcription" This is a grossly misleading statement. What the authors reported are two conformation states of the L-P complex, without the template or the product/substrate. The active machine for viral RNA synthesis needs to include the nucleocapsid (which is the template). The authors only reported two conformation states: In Lintegral, MTase is a new position, and a tunnel may allow the GpppA-RNA reaches its active site. But the same tunnel may allow the genomic RNA to exit during replication. Without GpppA-, the genomic RNA may simply bypass MTase. In Lbody, the authors simply do not see where CD-MTase-CTD is. This conformation may be unfavorable to transcription, but it may also be unfavorable to replication. How do they know?

"Coordination of replication and transcription) is dependent on many factors, e.g. N0, nucleocapsid, host factors, etc. It would not be just the conformation of CD-MTase-CTD. The authors can only claim that Lintegral is favorable for transcription, and Lbody is not. They have no say on replication.

2. "Parallel P tetramers in MuV L-P complex"

"POD core fragments (P249–271) of P1 and P4 were successfully docked, but the helices 348 of P2 and P3 failed to fit accurately into the density. We then inverted the orientation of the P2–P3 dimer, yielding a reasonable coordinate of the parallel P tetramer."

The fit of P2/P3 dimer seems to be arbitrary. Based on Extended Data Fig. 4b, the fit seems to be very poor, especially for P3. Almost no residue sidechains were fit into densities. One of the unique features that identifies the helix orientations is the kink at residue G246, which is not included in the structures reported here. Taking account of structural flexibility, anti-parallel orientation of P2/P3 cannot be ruled out. The published structural and functional data on mumps virus P strongly support the anti-parallel orientation. The authors must consider both possibilities.

Other points:

199 Diverse origins of L-binding PXD among nsNSVs provide direct evidence to both
200 cartwheeling and sliding models to describe the advance of polymerase on NC.

This is incorrect. One cannot generalize to all nsNSVs. P proteins are not always tetramers, not always parallel, in nsNSVs.

242 P is a requisite for RNA synthesis only in nsNSVs. This statement is too general. Some members of nsNSVs do not have a P protein.

144 6d). Taken together, two conformations including MuV L-integral-P and L-body-P may represent the
145 transcriptional and replicational states, respectively.

This statement is not supported by the results, as discussed above.

243 further self-assembles into anti-parallel tetramers in vitro^{36,41}

This cryoEM study is also in vitro. "further self-assembles into anti-parallel tetramers in case of recombinant POD."

This domain

128 rearrangement from the PIV5 mode to the MuV mode needs a 90° rotation of MTase-CTD, but
129 the rotation path is blocked by CD (Fig. 2b). The direct transition between the PIV5 and MuV
130 modes seems impossible.

Such a discussion is meaningless. No body expects such a rotation without dissociation of the MTase-CTD region first, before re-association with the L-body.

Their interactions

162 are majorly mediated by hydrogen bonds, salt bridges, and hydrophobic interactions.

All protein interactions are mediated by these interactions. This statement is not needed.

143 formed by RdRp and PRNTase is available, potentially as a replication state. PRNTase is not required for replication.

24 (CTD) adopt a distinct spatial organization from parainfluenza virus 5 (PIV5), " (CTD) adopt a spatial organization different from that of parainfluenza virus 5 (PIV5)"

28 model of MuV P helps build uncertain residues to the C-terminal regions "model of MuV P helps in building uncertain residues in the C-terminal regions".

Reviewer #2 (Remarks to the Author):

The viral RNA-dependent RNA polymerase (RdRp) complex is composed of large protein (L) and phosphoprotein (P) is responsible for the transcription and replication of viral genome RNA. Using cryo-EM, Li et al. presented two distinct structures of the L-P complex of the mumps virus, which may represent a transcription state and a replication state. The protein folding and the spatial configurations of the mumps virus L-P complex were similar to those observed in other members of the order Mononegavirales. The P tetramer was bound to the RdRp domain of L through its two P molecules. Based on the structure together with previously reported L-P structures of the other mononegaviruses, the authors concluded that a sliding model is preferable for explaining the mechanism by which the polymerase advances along the helical nucleocapsid.

Overall, the manuscript is clearly written, and the structural analysis is well-executed. On the other hand, some interpretations lack sufficient experimental support, and the reviewer feels that a clear distinction needs to be made between experimental and predicted structures and between results and discussion.

Major points:

1. Lines 124-132

The possibility that the L conformations differ between viral species cannot be ruled out. For this reviewer, it is more reasonable to assume that MuV and PIV5 show distinct conformations and that MuV cannot adopt the conformation of PIV5 type. Further validation, such as biochemistry or computational chemistry, is needed to conclude that there is a conformational transition.

2. Lines 133-145

The authors should experimentally evaluate if the two conformations represent the state of transcription and replication, because this is the most valuable finding in this study. This reviewer suggests performing some structural-base mutagenesis to validate this difference (e.g. in vitro RNA synthesis, strand-specific qPCR within infected or transfected cells, whatever). Without experimental validation, this is best left to a description in the Discussion section and the title must be changed.

3. Lines 153-158

The meaning of "parallel" is vague and confusing. The cryo-EM map of P, to which the atomic coordinates were not assigned, seems to show the twisting α -helices. The crystal structure (PDB-ID: 4EIJ) in Assembly 1 has a similar composition of the P tetramer to the cryo-EM structure, although the interface between the asymmetric units (P dimers) is different. The difference between cryo-EM (this study) and crystal structure (ref 36) of P and the similarity of P structures on L among the nsRNA viruses are also difficult to understand from the manuscript. The reviewer suggests that adding side-by-side comparison figures will help readers understand.

4. Line 339, DeepEMhancer

Although not explicitly stated in the text, it appears that the authors used the 3D volume modified by DeepEMhancer for the FSC calculations (resolution estimates), real-space refinement, model validation, figures in the manuscript, etc. Although it may be helpful in interpreting the Coulomb potential map calculated by single-particle cryo-EM, the 3D volume after AI-based map modifications is not experimental data. The author should distinguish and clearly state what was seen and validated from the experiments and what was suggested by the AI-based aid. The author should use the unmodified 3D map, such as a b-factor sharpened map, as the basis for results and figures, as the map for model refinement, and as the primary map in the PDB deposition, and include the 3D volume obtained by DeepEMhancer as an additional map.

5. Line 337 and Extended Data Fig. 2, "Composite"

The authors appear to have created a composite map (combined map) and used it as the final map for atomic modeling and primary map, which is inappropriate. How did the authors calculate the "gold standard" FSC? A composite map can be included in PDB deposition but is not an experimental structure and does not guarantee that the full-length L has the compositions of the locally refined subdomains. The locally refined map should be deposited in the PDB, and the atomic models should be built separately on each subdomain. For detailed rules and regulations, please refer to the following URL. "Can I deposit a composite map to EMDB?" <https://www.ebi.ac.uk/emdb/faq#a5>

Minor points:

6. Abbreviations

The reviewer suggests that the authors consider using the species names and abbreviations from the most recent ICTV Virus Metadata Resource, e.g., change "PIV5" to "PIV-5", "respiratory syncytial virus (RSV)" to "human respiratory syncytial virus (HRSV)", vesicular stomatitis virus (VSV) to vesicular stomatitis Indiana virus (VSIV). The others can be found at https://ictv.global/news/vmr_release_0423.

7. Figure 1d

The map of P-tetramers in Lbody-P is relatively obscure. If the map is obscured by heterogeneity in this interface, this reviewer suggests exploring alternative tetramer placements by conducting focused refinement only around this part. This approach may provide additional insights for considering the RNA synthesis mechanism, whether it is the sliding or the cartwheel model.

8. Clash scores of the atomic models

The clash scores of the atomic models appear to be quite high. This reviewer recommends refining the models with ISOLDE, which would help improve the clash score. Alternatively, carefully inspect the relevant clash areas in Coot. There is no need to stick to 'good values,' but the best possible model

based on your experimental cryo-EM map should be built and provided.

9. Line 247

"Zaire ebolavirus POD are trimers, whereas P forms tetramers in polymerase complexes" For filovirus P protein should be abbreviated as VP35.

10. Line 266

"100,000g" should be "100,000 × g"

11. 3. Lines 267, 275, 283, 298, 305, 307, 308, 309, 312

Need to add a space before °C.

1 **Reviewer #1 (Remarks to the Author):**

*The authors reported the cryoEM structure of the L-P complex of mumps virus. The*
*L-P complex is the viral RdRp responsible for viral RNA synthesis using the*
*nucleocapsid as the template. The structural results are important because mumps*
*virus is a re-emerging human pathogen. There are frequent outbreaks in recent years,*
*up to thousands of cases per year.*

*The primary finding of this work is that the viral RdRp of mumps virus has a very*
*similar structure as that in other important human pathogens, such as PIV5, RSV,*
*EBOV and RABV. This confirms that antiviral approaches to other nsNSVs are also*
*applicable to MuV.*

*However, the presentation of their results suffers from overclaiming the significance.*
*Many statements are not based on the actual observations.*

**Re:**

**Thanks for reviewing our manuscript.**

**The mumps virus (MuV) is a highly contagious human pathogen and frequently causes**
**worldwide outbreaks despite available vaccines. So far, there are no anti-viral**
**medications that can treat mumps. Via cryo-EM, we resolved two conformations of**
**MuV L-P complex: $L_{\text{body}}\text{-P}$ and $L_{\text{integral}}\text{-P}$. In both conformations, their core domains**
**including RdRp and PRNTase are conserved in structure with other important human**
**pathogens, such as PIV-5, HRSV, EBOV and RABV, which indicates that antiviral**
**approaches developed for these nsNSVs are also applicable to MuV.**

**We agree with the reviewer that we overclaimed the significance especially on the**
**coordination of genome replication and transcription. We have followed the advice and**
**rephrased the related descriptions.**

**Thanks.**

*1. "Coordination of replication and transcription" This is a grossly misleading*
*statement. What the authors reported are two conformation states of the L-P complex,*
*without the template or the product/substrate. The active machine for viral RNA*
*synthesis needs to include the nucleocapsid (which is the template). The authors only*
*reported two conformation states: In L_{integral} , MTase is a new position, and a tunnel*
*may allow the GpppA-RNA reaches its active site. But the same tunnel may allow the*
*genomic RNA to exit during replication. Without GpppA-, the genomic RNA may*
*simply bypass MTase. In L_{body} , the authors simply do not see where CD-MTase-CTD*
*is. This conformation may be unfavorable to transcription, but it may also be*
*unfavorable to replication. How do they know?*

*"Coordination of replication and transcription) is dependent on many factors, e.g.*
*N0, nucleocapsid, host factors, etc. It would not be just the conformation of CD-*
*MTase-CTD. The authors can only claim that L_{integral} is favorable for transcription,*
*and L_{body} is not. They have no say on replication.*

**Re:**

**Thanks for the great comments.**

We totally agreed with the reviewer that it is improper for us to use “coordination of
replication and transcription”, which overclaimed the significance of our structures. We
remodeled our research aim to resolve structures of MuV L–P corresponding to genome
replication and transcription via cryo-electron microscopy.

We followed the reviewer’s advice and only claimed that MuV L_{integral}–P is favorable
for transcription due to its continuous tunnel to the MTase domain. MuV L_{body}–P owns
a flexible appendage, and its RNA tunnel is inaccessible to MTase-CTD, which is
deemed unfavorable as a transcription state. However, the RNA cavity formed by RdRp
and PRNTase domains is still available, with potential capability of genome replication.
Only in the Discussion section, we discussed the possibility of MuV L_{integral}–P, MuV
L_{body}–P, and PIV-5 L–P as the replication state.

Thanks.

2. “Parallel P tetramers in MuV L–P complex”

*“POD core fragments (P249–271) of P1 and P4 were successfully docked, but the
helices 348 of P2 and P3 failed to fit accurately into the density. We then inverted the
orientation of the P2–349 P3 dimer, yielding a reasonable coordinate of the parallel
P tetramer.”*

*The fit of P2/P3 dimer seems to be arbitrary. Based on Extended Data Fig. 4b, the fit
seems to be very poor, especially for P3. Almost no residue sidechains were fit into
densities. One of the unique features that identifies the helix orientations is the kink at
residue G246, which is not included in the structures reported here. Taking account
of structural flexibility, anti-parallel orientation of P2/P3 cannot be ruled out. The
published structural and functional data on mumps virus P strongly support the anti-
parallel orientation. The authors must consider both possibilities.*

Re:

Thanks for the great comments.

In both MuV L_{body}–P and L_{integral}–P, four P molecules assemble like a kettle spout
anchored to L. Due to the structural flexibility, only part of P_{OD} tetramer was resolved,
which was hard to distinguish the orientations of P1–P4. We followed the reviewers’
advice, and performed local refinements on P tetramer after recentering (*Extended data
Fig. 2*). This helps us resolve the full P_{OD} tetramer at the resolution of 3.49 Å. We
docking P1 and P4 into the cryo-EM map, and the fitting is perfect. However, we still
could not recognize the kink at residues Gly₂₄₆ in P2/P3. We attempted to dock P2/P3
in either a parallel or anti-parallel manner, and the parallel fitting of P2/P3 is better than
the anti-parallel one (*Extended data Fig. 4b,c*). We also checked the EM maps of P
tetramers from other nsNSVs and their respective atomic models. EBOV VP35
tetramers were well resolved (EMD-33775, PDB ID 7YER), and clearly adopted a
parallel manner (*Yuan et al. Nature 2022, PMID 36171293*). Combining all these
information, we prefer P tetramer in a parallel manner. Certainly, we also clearly
pointed out the other possibility of docking P2/P3 in an anti-parallel way in the
Discussion section.

Our further analyses on parallel P tetramers reveal the diverse origins of L-binding P_{XD}
in nsNSVs, which contradicts with the fixed origin of L-binding P_{XD} proposed in the
cartwheeling model. Apparently, anti-parallel MuV P tetramers do not support the
cartwheeling model for L–P complex, either. All these strongly point to a sliding model
that any P_{XD} in tetrameric P can stably bind to RdRp, and other P_{XD} will reengage with
the RdRp domain only after the falling-off of the current P_{XD} from L.

Thanks.

*Other points:*

*Diverse origins of L-binding P_{XD} among nsNSVs provide direct evidence to both*
*cartwheeling and sliding models to describe the advance of polymerase on NC.*
*This is incorrect. One cannot generalize to all nsNSVs. P proteins are not always*
*tetramers, not always parallel, in nsNSVs.*

Re:

Thanks for the great comment.

We totally agreed with the reviewer that P has varying assembly forms. Nipah virus
P_{OD} assembles into trimers in solution, but is crystalized into tetramer. Crytal structures
of Zaire ebolavirus VP35 oligomerization domain are trimers, whereas VP35 forms
tetramers in polymerase complexes. We have talked about the different assembly forms
of P_{OD} in our Discussion section. Accordingly, we have deleted our improper
description.

Thanks.

*P is a requisite for RNA synthesis only in nsNSVs. This statement is too general. Some*
*members of nsNSVs do not have a P protein.*

Re:

Thanks for pointing out the improper description.

We followed the reviewer’s advice and rephrased the statement to “P is required for
RNA synthesis in most nsNSVs”.

Thanks.

*Line 144: Taken together, two conformations including MuV L_{integral}–P and L_{body}–P*
*may represent the transcriptional and replicational states, respectively.*
*This statement is not supported by the results, as discussed above.*

Re:

Thanks again for the great comment.

We have followed the advice and deleted the improper statements. Meanwhile, we
rephrased our research aim to structural determination of MuV L–P.

Thanks.

*Line 243: further self-assembles into anti-parallel tetramers in vitro. This cryoEM*
*study is also in vitro. “further self-assembles into anti-parallel tetramers in case of*
*recombinant POD.”*

Re:

Thanks for pointing out our improper description.

We have followed the advice and rephrased the sentence as “MuV P itself forms parallel
dimers and further self-assembles into anti-parallel tetramers in case of recombinant
POD”.

Thanks.

*This domain rearrangement from the PIV5 mode to the MuV mode needs a 90°*
*rotation of MTase-CTD, but the rotation path is blocked by CD (Fig. 2b). The direct*
*transition between the PIV5 and MuV modes seems impossible.*
*Such a discussion is meaningless. No body expects such a rotation without*
*dissociation of the MTase-CTD region first, before re-association with the L-body.*

Re:

Thanks for pointing out our improper description.

We agreed with the reviewers that L conformations differ among different viral species,
especially on the spatial organization of CD-MTase-CTD. We performed deep 3D
classification using PIV-5 and MuV structures as the multiple references, and only
MuV-like structure is resolved (*Rebuttal Fig. 1*). Thus, only one L conformation is
resolved from each viral species, which excludes the possibility of structural transition
between the PIV-5 and MuV modes via rotating MTase-CTD.

We followed the reviewer’s suggestion and deleted the related description.

Thanks.

Rebuttal Fig. 1 Multiple-reference 3D classification of MuV L–P complex. **a**, MuV $L_{\text{integral-P}}$, $L_{\text{body-P}}$, and PIV-5 structures were duplicated one time, and utilized as the references for 3D classification on the whole dataset. **b**, MuV $L_{\text{integral-P}}$ and PIV-5 structures were duplicated one time, and utilized as the references for 3D classification on the $L_{\text{integral-P}}$ particles. No PIV-5-like structures can be resolved from MuV L–P complex.

Their interactions are majorly mediated by hydrogen bonds, salt bridges, and hydrophobic interactions. All protein interactions are mediated by these interactions. This statement is not needed.

Re:

Thanks for pointing out the improper description.

We followed the advice and deleted the statement.

Thanks.

formed by RdRp and PRNTase is available, potentially as a replication state. PRNTase is not required for replication.

Re:

Thanks for pointing out the confusing description.

To make it clear, we rephrased our description as “the RNA cavity formed by RdRp and PRNTase domains is available, with potential capability for genome replication”.

Thanks.

*(CTD) adopt a distinct spatial organization from parainfluenza virus 5 (PIV5),*
*“(CTD) adopt a spatial organization different from that of parainfluenza virus 5*

**Re:**

**Thanks for pointing out the improper description.**

**We have followed the advice and rephrased the sentence accordingly.**

**Thanks.**

*(PIV5)” model of MuV P helps build uncertain residues to the C-terminal regions*
*“model of MuV P helps in building uncertain residues in the C-terminal regions”.*

**Re:**

**Thanks for pointing out the improper description.**

**We followed the advice and revised the sentence.**

**Thanks.**

**Reviewer #2 (Remarks to the Author):**

*The viral RNA-dependent RNA polymerase (RdRp) complex is composed of large*
*protein (L) and phosphoprotein (P) is responsible for the transcription and*
*replication of viral genome RNA. Using cryo-EM, Li et al. presented two distinct*
*structures of the L-P complex of the mumps virus, which may represent a*
*transcription state and a replication state. The protein folding and the spatial*
*configurations of the mumps virus L-P complex were similar to those observed in*
*other members of the order Mononegavirales. The P tetramer was bound to the RdRp*
*domain of L through its two P molecules. Based on the structure together with*
*previously reported L-P structures of the other mononegaviruses, the authors*
*concluded that a sliding model is preferable for explaining the mechanism by which*
*the polymerase advances along the helical nucleocapsid.*
*Overall, the manuscript is clearly written, and the structural analysis is well-*
*executed. On the other hand, some interpretations lack sufficient experimental*
*support, and the reviewer feels that a clear distinction needs to be made between*
*experimental and predicted structures and between results and discussion.*

**Re:**

**Thanks for reviewing our manuscript.**

**We totally agreed with the reviewer that some statements lack sufficient experimental**
**support. Just as what the reviewer suggested, we made a clear distinction between**
**experimental and predicted structures. We also followed the advice and moved the**
**speculations to the Discussion section.**

**Thanks.**

*Major points:*
*1. Lines 124-132*
*The possibility that the L conformations differ between viral species cannot be ruled*
*out. For this reviewer, it is more reasonable to assume that MuV and PIV5 show*
*distinct conformations and that MuV cannot adopt the conformation of PIV5 type.*
*Further validation, such as biochemistry or computational chemistry, is needed to*
*conclude that there is a conformational transition.*

**Re:**

**Thanks for the great comment.**

**We agreed with the reviewers that L conformations differ among different viral species**
**on the spatial organization of CD-MTase-CTD. We performed deep 3D classification**
**on MuV L-P complex using the PIV-5 and MuV structures as the references, and only**
**MuV-like conformation can be resolved (Rebuttal Fig. 1). All these verified the**
**reviewer's speculation that MuV and PIV-5 may possess distinct conformations. Thus,**
**we removed the description about the conformational transition between PIV-5-like and**
**MuV L_{integral}-P-like structure from the manuscript.**

Thanks.

2. Lines 133-145

*The authors should experimentally evaluate if the two conformations represent the*
*state of transcription and replication, because this is the most valuable finding in this*
*study. This reviewer suggests performing some structural-base mutagenesis to*
*validate this difference (e.g. in vitro RNA synthesis, strand-specific qPCR within*
*infected or transfected cells, whatever). Without experimental validation, this is best*
*left to a description in the Discussion section and the title must be changed.*

Re:

Thanks for the great comment.

nsNSVs L–P complexes catalyze RNA synthesis in both genome replication and
transcription. As the core component of L–P complex, one L structure is usually
resolved from each viral species, but L differs in structure among different species,
especially on the spatial organization of CD-MTase-CTD, which makes it hard to link
conformations of L–P complexes with replication and transcription activities. Via cryo-
EM, we resolved two conformations of the MuV L–P complex: L_{body}–P and L_{integral}–P.
Interestingly, L_{integral}–P possesses a continuous RNA tunnel to the MTase domain
preferable as the transcription state, while L_{body}–P takes an appendage-free
conformation, unfavorable for transcription. Considering that L_{body}–P still owns an
exposed RNA cavity formed by RdRp and PRNTase domains, with potential capability
for genome replication.

We performed *in vitro* RNA synthesis assay via incubating the radio-labelled RNA
sequence with purified L–P complex. The gels clearly showed the catalytic activity of
MuV L–P as the RNA-dependent RNA polymerase. Unfortunately, our RNA synthesis
assay was unable to distinguish the replication and transcription activities. We checked
the recently-published papers on nsNSVs L–P (Yuan *et al. Nature* 2022, PMID
36171293; Cong *et al. Nat commun* 2023, PMID 36898997), and it seems still immature
to clearly distinguish these two activities. We are still working to pave a way to combine
both mutagenesis and functional assays to fully address this issue.

At the current stage, we intend to change our title to “Structures of the mumps virus
polymerase complex via cryo-electron microscopy” and talk about the functions of L–
P complex in the Discussion section.

Thanks.

3. Lines 153-158

*The meaning of "parallel" is vague and confusing. The cryo-EM map of P, to which*
*the atomic coordinates were not assigned, seems to show the twisting α -helices. The*
*crystal structure (PDB-ID: 4EIJ) in Assembly 1 has a similar composition of the P*
*tetramer to the cryo-EM structure, although the interface between the asymmetric*
*units (P dimers) is different. The difference between cryo-EM (this study) and crystal*
*structure (ref 36) of P and the similarity of P structures on L among the nsRNA*

*viruses are also difficult to understand from the manuscript. The reviewer suggests*
*that adding side-by-side comparison figures will help readers understand.*

**Re:**

Thanks for the great comment.

The crystal structure (PDB ID 4EIJ) of MuV P revealed an anti-parallel tetramer
(parallel dimers in anti-parallel configuration). Specifically, P1 and P4 form a parallel
dimer, and P2 and P3 form another. In Assembly 1, P1/P4 dimer and P2/P3 dimer
further assemble in an anti-parallel manner. This kind of configuration of MuV P_{OD} is
named as anti-parallel. Different from crystal structures of MuV P tetramer, P1/P4
dimer and P2/P3 dimer pack in a parallel way in many other nsNSVs L–P complexes,
which is called a parallel manner.

In MuV L–P complex, four P molecules assemble like a kettle spout anchored to L. Due
to the structural flexibility, only part of P_{OD} oligomers is resolved. We followed the
reviewer's advice, and performed local refinements on P after recentering (*Fig. 1b–d*
and *Extended data Figs. 2–4*). This operation helped us resolve the full P_{OD} oligomers.
We docked P1 and P4 into the cryo-EM densities, and the fitting is perfect. However,
we still could not recognize the kink at residues Gly₂₄₆, which is critical to recognize
the orientation of P2/P3. We attempted to dock P2/P3 in either parallel or anti-parallel
manner, and parallel P2/P3 fit better than anti-parallel P2/P3 (*Extended data Fig. 4b,c*).

To avoid the confusion, we followed the reviewer's advice and supplied side-by-side
comparison figures (*Extended data Fig. 4b,c*) for better illustration.

**Thanks.**

**4. Line 339, DeepEMhancer**

*Although not explicitly stated in the text, it appears that the authors used the 3D*
*volume modified by DeepEMhancer for the FSC calculations (resolution estimates),*
*real-space refinement, model validation, figures in the manuscript, etc. Although it*
*may be helpful in interpreting the Coulomb potential map calculated by single-*
*particle cryo-EM, the 3D volume after AI-based map modifications is not*
*experimental data. The author should distinguish and clearly state what was seen and*
*validated from the experiments and what was suggested by the AI-based aid. The*
*author should use the unmodified 3D map, such as a b-factor sharpened map, as the*
*basis for results and figures, as the map for model refinement, and as the primary*
*map in the PDB deposition, and include the 3D volume obtained by DeepEMhancer*
*as an additional map.*

**Re:**

Thanks for the great comment.

DeepEMhancer is a deep learning model on pairs of experimental volumes and atomic-
corrected volumes, which can perform automatic sharpening of unmasked, unfiltered
reconstructions. Compared with global B-factor correction, DeepEMhancer can
improve the map local quality, and is widely utilized for better interpretability (*Zhao et*

*al. Nature 2023, PMID 37524305; Metcalfe1 et al. Nat Commun 2023, PMID*
*37558661; Bodrug et al. NSMB 2023, PMID 37735619; Orta et al. Science 2023, PMID*
*37440661).*

We followed the reviewer's advice and clearly pointed out that which maps are
sharpened by DeepEMhancer in the Methods section and in the figure legends, and
clearly distinguish the sharpened EM maps and raw data maps. Meanwhile, we
uploaded both unsharpened and DeepEMhancer sharpened maps for each part of MuV
$L_{\text{integral-P}}$ and $L_{\text{body-P}}$, and details are listed in *Supplementary Tables 1 and 2*.

Thanks.

*5. Line 337 and Extended Data Fig. 2, "Composite"*

*The authors appear to have created a composite map (combined map) and used it as*
*the final map for atomic modeling and primary map, which is inappropriate. How did*
*the authors calculate the "gold standard" FSC? A composite map can be included in*
*PDB deposition but is not an experimental structure and does not guarantee that the*
*full-length L has the compositions of the locally refined subdomains. The locally*
*refined map should be deposited in the PDB, and the atomic models should be built*
*separately on each subdomain. For detailed rules and regulations, please refer to the*
*following URL. "Can I deposit a composite map to EMDB?"*
*<https://www.ebi.ac.uk/emdb/faq#a5>*

Re:

Thanks for the great suggestion.

We performed 3D reconstruction on MuV L-P complex as the whole, and obtained a
cryo-EM map at the resolution of 3.02 Å (*Extended data Fig. 3a*), based on the gold-
standard FSC at the criterion of 0.143. In MuV L-P complex, RdRp and PRNTase
domains are much better resolved than CD-MTase-CTD and P. To improve the
resolutions of these two flexible regions, we performed local refinements on RdRp-
PRNTase, CD-MTase-CTD, and P, separately, and improved their respective local
resolutions to 2.93, 3.13, and 3.49 Å (*Extended data Fig. 3b-d*).

We noticed that the resolutions of RdRp-PRNTase, CD-MTase-CTD, and P still vary a
lot. For better interpretability, we built a composite map of $L_{\text{integral-P}}$ and $L_{\text{body-P}}$ in
Phenix. We followed the reviewer's advice and uploaded both unsharpened and
DeepEMhancer sharpened maps for each part of $L_{\text{integral-P}}$ and $L_{\text{body-P}}$ to EMDB.
Additionally, we uploaded the composite maps of $L_{\text{integral-P}}$ and $L_{\text{body-P}}$ as a
supplement in the EMDB uploading system, as indicated in other papers (*Afsar et al.*
*Nat Commun 2023, PMID 37553340; Chen et al. NSMB 2023, PMID 37932450; Zhao*
*et al, Sci Adv 2023, PMID 37595043; Zhang et al. Science 2023, PMID 37384673). We*
clearly pointed out that which maps are composite maps after DeepEMhancer
sharpening in the figure legends.

Thanks.

*Minor points:*

6. Abbreviations

*The reviewer suggests that the authors consider using the species names and*
*abbreviations from the most recent ICTV Virus Metadata Resource, e.g., change*
*“PIV5” to “PIV-5”, “respiratory syncytial virus (RSV)” to “human respiratory*
*syncytial virus (HRSV), vesicular stomatitis virus (VSV) to vesicular stomatitis*
*Indiana virus (VSIV). The others can be found at*
*https://ictv.global/news/vmr_release_0423.*

Re: Thanks so much for pointing out the abbreviation issue.

We have followed the advice and corrected all the abbreviations throughout the
manuscript including main text, figures, and supplementary materials.

Thanks.

7. Figure 1d
*The map of P-tetramers in L_{body}-P is relatively obscure. If the map is obscured by*
*heterogeneity in this interface, this reviewer suggests exploring alternative tetramer*
*placements by conducting focused refinement only around this part. This approach*
*may provide additional insights for considering the RNA synthesis mechanism,*
*whether it is the sliding or the cartwheel model.*

Re:

Thanks for the great comment.

We tried the focal refinement on P tetramers in L_{integral}-P and L_{body}-P. Unfortunately,
we could not optimize the local resolutions. As suggested by the reviewer, we moved
the particles centers from L to P tetramer, and obtained more cryo-EM densities in the
map. Based on the better-resolved maps, we tried different docking of P tetramer. We
are very confident on the positions of P1 and P4. Unfortunately, we could not clearly
point out the orientation of P2 and P3. Based on the published results and sequence
conservancy, we preferred the parallel model. In the Discussion section, we still
emphasized the other possibility of P tetramer packing in an anti-parallel manner.

Thanks.

8. Clash scores of the atomic models
*The clash scores of the atomic models appear to be quite high. This reviewer*
*recommends refining the models with ISOLDE, which would help improve the clash*
*score. Alternatively, carefully inspect the relevant clash areas in Coot. There is no*
*need to stick to 'good values,' but the best possible model based on your experimental*
*cryo-EM map should be built and provided.*

Re:

Thanks for the great comments.

We performed local refinement on P tetramer after recentering, and obtained a new
cryo-EM maps. Accordingly, we optimized the atomic model, and the clash score

dropped to ~19 in PHENIX (18 in PDB validation report).

Thanks.

9. Line 247

*“Zaire ebolavirus POD are trimers, whereas P forms tetramers in polymerase*
*complexes” For filovirus P protein should be abbreviated as VP35.*

Re:

Thanks for pointing out the improper description.

We followed the advice and changed the P to VP35 in EBOV.

Thanks.

10. Line 266

*“100,000g” should be “100,000 × g”*

Re:

Thanks for pointing out the improper usage.

We followed the advice and revised our manuscript.

Thanks.

11. 3. Lines 267, 275, 283, 298, 305, 307, 308, 309, 312
*Need to add a space before °C.*

Re:

Thanks so much for pointing out the improper usage.

We followed the advice and added a space before “°C” throughout the manuscript.

Thanks.

REVIEWER COMMENTS

Reviewer #1 (Remarks to the Author):

Most of the comments from this reviewer were well addressed. Here is another suggestion:

"This observation is highly consistent with P molecules in other nsNSVs21-24,26,34,58-60, 159 which indicates a conserved mechanism for P molecules to mediate RNA genome replication and 160 transcription."

Not all other nsNSVs have parallel P tetramers. P in VSIV and RABV are dimers. P has many ways (different in each virus) to mediate RNA transcription and replication. So be more precise:

"This observation is highly consistent with P molecules in many other nsNSVs21-24,26,34,58-60, 159 which indicates a generally conserved mechanism for P molecules to mediate RNA genome replication and 160 transcription."

Reviewer #2 (Remarks to the Author):

Our suggestions regarding virological aspects have been addressed, but the proper handling of structural information remains unsatisfactory.

Major Comments

Nevertheless, authors should ensure that they use maps appropriately for their respective purposes and clearly describe which maps were used for each figure, composite map creation, model refinement, etc.

DeepEMhancer does not "sharpen" the map or "improve a quality" but modifies it to look more like proteins, e.g., without natural noise patterns. The training data set of this program should not contain any water, ions, nucleic acid, etc. other than the atoms that compose proteins, although the model usually does not suffer from incorrect geometry when using low-resolution maps.

Composite maps are also not experimental data for atomic modeling, because the existence of their specific pose has not been experimentally validated, and the boundary area of the original maps may be inaccurate.

Judging from the file names of the maps and models and our actual inspection of them, the models appear to be built on composite maps created from two maps modified with DeepEMhancer. In fact, the model has an unnatural geometry, especially in the side chains at the boundary region of L and P and in both atomic models.

The fact that they are used for model refinement in many papers does not guarantee their scientific validity, and how they are used is critical. In fact, an increasing number of studies use modified maps (e.g., maps after AI-based modifications, composite maps) without proper validation.

These issues have been widely discussed in scientific communities such as CCPEM to reach a better consensus.

<https://www.jiscmail.ac.uk/cgi-bin/wa-jisc.exe?A2=ind2310&L=CCPEM&O=D&P=67154>

Minor Comments

Expanded Figure 11, It should be easier for the reader to understand by specifying what each virus abbreviation stands for. In addition, the species names are still not up to date. For example, "SeV" should be "SenV", "NDV" should now be "avian paramyxovirus 1 (APMV-1)".

Line 563, "EMD-37964 (P of Lbody-P)" should be "EMD-37962 (P of Lbody-P)"

**Reviewer #1 (Remarks to the Author):**

*Most of the comments from this reviewer were well addressed. Here is another*
*suggestion:*

*"This observation is highly consistent with P molecules in other nsNSVs which*
*indicates a conserved mechanism for P molecules to mediate RNA genome replication*
*and transcription."*

*Not all other nsNSVs have parallel P tetramers. P in VSIV and RABV are dimers. P*
*has many ways (different in each virus) to mediate RNA transcription and replication.*
*So be more precise:*

*"This observation is highly consistent with P molecules in many other nsNSVs which*
*indicates a generally conserved mechanism for P molecules to mediate RNA genome*
*replication and transcription."*

**Re:**

**Thanks again for reviewing our manuscript.**

**We have followed the reviewer's great comment and rephrased the sentence**
**accordingly.**

**Again, thanks a lot for your time on our manuscript.**

**Reviewer #2 (Remarks to the Author):**

*Our suggestions regarding virological aspects have been addressed, but the proper*
*handling of structural information remains unsatisfactory.*

*Major Comments*

*Nevertheless, authors should ensure that they use maps appropriately for their*
*respective purposes and clearly describe which maps were used for each figure,*
*composite map creation, model refinement, etc.*

*DeepEMhancer does not "sharpen" the map or "improve a quality" but modifies it to*
*look more like proteins, e.g., without natural noise patterns. The training data set of*
*this program should not contain any water, ions, nucleic acid, etc. other than the*
*atoms that compose proteins, although the model usually does not suffer from*
*incorrect geometry when using low-resolution maps.*

*Composite maps are also not experimental data for atomic modeling, because the*
*existence of their specific pose has not been experimentally validated, and the*
*boundary area of the original maps may be inaccurate.*

*Judging from the file names of the maps and models and our actual inspection of*
*them, the models appear to be built on composite maps created from two maps*
*modified with DeepEMhancer. In fact, the model has an unnatural geometry,*
*especially in the side chains at the boundary region of L and P and in both atomic*
*models.*

*The fact that they are used for model refinement in many papers does not guarantee*
*their scientific validity, and how they are used is critical. In fact, an increasing*
*number of studies use modified maps (e.g., maps after AI-based modifications,*
*composite maps) without proper validation.*

*These issues have been widely discussed in scientific communities such as CCPEM to*
*reach a better consensus.*

*[https://www.jiscmail.ac.uk/cgi-bin/wa-](https://www.jiscmail.ac.uk/cgi-bin/wa-jisc.exe?A2=ind2310&L=CCPEM&O=D&P=67154)*
*[jisc.exe?A2=ind2310&L=CCPEM&O=D&P=67154](https://www.jiscmail.ac.uk/cgi-bin/wa-jisc.exe?A2=ind2310&L=CCPEM&O=D&P=67154)*

**Re:**

**Thanks so much for pointing out our improper use of the composite maps and models**
**refined from DeepEMhancer maps.**

**We really appreciated the link provided by the reviewer about the composite maps and**
**models refined from DeepEMhancer maps. Based on the discussion in the field, it is**
**more reasonable to refine models against the B-factor sharpened maps instead of**
**DeepEMhancer processed maps. We followed the instruction and refined models of**
**both MuV L_{integral}-P and L_{body}-P against the B-factor sharpened maps. Overall, the new**
**models are almost identical to our previous ones on the main chains. After refinements**
**against B-factor sharpened maps, the side chains in the new models improve slightly**
**on the specs such as clash score and rotamer.**

**We have followed the reviewer's great suggestion and done the following**
**improvements:**

**(1) We clearly pointed out the origin of maps in each figure in the figure legend**

of Fig. 1 (lines 589-593 in Article File).

(2) We clearly described the relationship of the composite maps and each
individual maps in the *Methods* section, the *Data Availability* section, and
*Extended Data Fig. 2* (lines 348-355, 365-366, and 379-389 in Article File).

(3) We uploaded the B-factor sharpened composite maps of MuV L_{integral}-P and
L_{body}-P as the additional maps to EM Data Bank (lines 379-389 in Article File).

(4) We updated the models of MuV L_{integral}-P and L_{body}-P in Protein Data Bank,
and *Supplementary Tables 1 and 2* (uploaded as the *Supplementary files: PDB-*
*8IZL.pdb and PDB-8X01.pdb*).

(5) For side-chain accuracy, we replotted *Fig. 3b-e* and *Extended Data Fig. 4a-b*
using the new models.

In all, we generated composite maps of MuV L_{integral}-P and L_{body}-P from their
respective locally refined maps. The composite maps of MuV L_{integral}-P and L_{body}-P
were either B-factor sharpened for atomic model refinement, or post-processed using
the DeepEMhancer to improve their interpretability for figure preparation.

Thanks.

*Minor Comments*

*Expanded Figure 11, It should be easier for the reader to understand by specifying*
*what each virus abbreviation stands for. In addition, the species names are still not up*
*to date. For example, "SeV" should be "SenV", "NDV" should now be "avian*
*paramyxovirus 1 (APMV-1)".*

Re:

Thanks for the comment.

We have followed the great advice and remodeled the *Extended Data Fig. 11* with the
species abbreviations from the most recent ICTV Virus Metadata Resource. We listed
the full name for each virus in the figure legend.

Thanks.

*Line 563, "EMD-37964 (P of L_{body}-P)" should be "EMD-37962 (P of L_{body}-*

*P)"* Re:

Thanks so much for pointing out our mistake.

We have corrected the EMDB ID of P of L_{body}-P to EMD-37962 (line 385 in Article
*File*).

Thanks so much for your hard work to improve our manuscript.

REVIEWERS' COMMENTS

Reviewer #2 (Remarks to the Author):

The authors have not yet addressed the points raised since the first peer review regarding the treatment of composite maps.

The review thinks that composite maps should not be used for atomic model modeling.

This point has also been discussed on the CCPEM mailing list, to which this reviewer has referred, and in Nakane et al.. <https://doi.org/10.7554/eLife.36861>

"...we note that this representation of a single atomic model for the entire complex is in principle not supported by the data. Besides creating a false impression of structural homogeneity, in particular, the conformations of residues at the interfaces of the rigid-body fitted atomic models may not reflect the true interface with the relative orientation of the bodies observed in the combined model."

Here, this reviewer provides the comments again with highlights.

First round

5. Line 337 and Extended Data Fig. 2, "Composite"

The authors appear to have created a composite map (combined map) and used it as the final map for atomic modeling and primary map, which is inappropriate. How did the authors calculate the "gold standard" FSC? A composite map can be included in PDB deposition but is not an experimental structure and does not guarantee that the full-length L has the compositions of the locally refined subdomains. The locally refined map should be deposited in the PDB, and the atomic models should be built separately on each subdomain. For detailed rules and regulations, please refer to the following URL. "Can I deposit a composite map to EMDB?" <https://www.ebi.ac.uk/emdb/faq#a5>

Second round

Composite maps are also not experimental data for atomic modeling, because the existence of their specific pose has not been experimentally validated, and the boundary area of the original maps may be inaccurate.

Reviewer #3 (Remarks to the Author):

The virological aspects of this paper appear well thought out and well reviewed by both reviewer #1 and reviewer #2, and on the scientific merit seems worthy of publication. The remaining discussion revolves around which maps were used for atomic model building.

Previously the authors were using DeepEMhancer enhanced composite maps for their model building. I agree with reviewer #2 that this is completely inappropriate. The authors are now model-building into b-factor sharpened composite maps. Reviewer #2 would clearly argue that this is inappropriate. The composite map does not exist in a real sense, and in fact particularly at the interface between different maps might well be artifactual leading to incorrect artifactual models.

Ideally, the authors would build their models into separate focused refined maps, then combine those models by fitting them into the the original non-composite map as their final atomic model. I believe if they could do this it would address everyone's concerns.

Having said that, some people do model build into their composite maps, and the field is not in complete agreement as to whether this is appropriate in some cases or not (though I lean towards it being inappropriate). In this case, I believe it should be for the authors to decide (and be judged)

what they want to do. I hope they can individually model into the distinct focused refined maps, and describe this in their methods.

If they they wish to continue with their current approach - one thing is clear, they must specifically mention this in the methods. Currently the methods state "The final coordinates of the L-P complexes were real-space refined against B-factor sharpened maps in PHENIX 1.20.1" This should at the very least be changed to explicitly state they refined against composite maps "refined against B-factor sharpened composite maps", and ideally they would also include a justification as to why they did this rather than refining into the individual focused refined maps.

**Reviewer #2 (Remarks to the Author):**

*The authors have not yet addressed the points raised since the first peer review*
*regarding the treatment of composite maps.*

*The review thinks that composite maps should not be used for atomic model modeling.*

*This point has also been discussed on the CCPEM mailing list, to which this reviewer*
*has referred, and in Nakane et al.. <https://doi.org/10.7554/eLife.36861>*

*“...we note that this representation of a single atomic model for the entire complex is*
*in principle not supported by the data. Besides creating a false impression of*
*structural homogeneity, in particular, the conformations of residues at the interfaces*
*of the rigid-body fitted atomic models may not reflect the true interface with the*
*relative orientation of the bodies observed in the combined model.”*

*Here, this reviewer provides the comments again with highlights.*

*First round*

*5. Line 337 and Extended Data Fig. 2, “Composite”*

*The authors appear to have created a composite map (combined map) and used it as*
*the final map for atomic modeling and primary map, which is inappropriate. How did*
*the authors calculate the “gold standard” FSC? A composite map can be included in*
*PDB deposition but is not an experimental structure and does not guarantee that the*
*full-length L has the compositions of the locally refined subdomains. The locally*
*refined map should be deposited in the PDB, and the atomic models should be built*
*separately on each subdomain. For detailed rules and regulations, please refer to the*
*following URL. “Can I deposit a composite map to EMDB?”*
*<https://www.ebi.ac.uk/emdb/faq#a5>*

*Second round*

*Composite maps are also not experimental data for atomic modeling, because the*
*existence of their specific pose has not been experimentally validated, and the*
*boundary area of the original maps may be inaccurate.*

**Re:**

**Thanks so much for your reviewing our manuscript.**

**As to the atomic model modeling, we followed your excellent instructions, refining**
**atomic models against each locally refined map and performing rigid body docking of**
**atomic models into the composite maps with the assistance of the globally refined maps.**
**In the revised manuscript, we have clearly mentioned the procedures for data processing**
**and model building.**

**We really appreciate your great effort in avoiding any possible mistakes in our**
**manuscript.**

**Thanks.**

**Reviewer #3 (Remarks to the Author):**

*The virological aspects of this paper appear well thought out and well reviewed by*
*both reviewer #1 and reviewer #2, and on the scientific merit seems worthy of*
*publication. The remaining discussion revolves around which maps were used for*
*atomic model building.*

*Previously the authors were using DeepEMhancer enhanced composite maps for their*
*model building. I agree with reviewer #2 that this is completely inappropriate. The*
*authors are now model-building into b-factor sharpened composite maps. Reviewer*
*#2 would clearly argue that this is inappropriate. The composite map does not exist in*
*a real sense, and in fact particularly at the interface between different maps might*
*well be artifactual leading to incorrect artifactual models.*

*Ideally, the authors would build their models into separate focused refined maps, then*
*combine those models by fitting them into the original non-composite map as their*
*final atomic model. I believe if they could do this it would address everyone's*
*concerns.*

*Having said that, some people do model build into their composite maps, and the field*
*is not in complete agreement as to whether this is appropriate in some cases or not*
*(though I lean towards it being inappropriate). In this case, I believe it should be for*
*the authors to decide (and be judged) what they want to do. I hope they can*
*individually model into the distinct focused refined maps, and describe this in their*
*methods.*

*If they wish to continue with their current approach - one thing is clear, they must*
*specifically mention this in the methods. Currently the methods state "The final*
*coordinates of the L-P complexes were real-space refined against B-factor sharpened*
*maps in PHENIX 1.20.1" This should at the very least be changed to explicitly state*
*they refined against composite maps "refined against B-factor sharpened composite*
*maps", and ideally they would also include a justification as to why they did this*
*rather than refining into the individual focused refined maps.*

**Re:**

**Thanks so much for reviewing our manuscript.**

**We have followed your great advice and built the individual models from distinct,**
**focused, refined maps. Meanwhile, we performed a rigid-body docking of individual**
**atomic models into the composite maps with the assistance of the globally refined maps.**

**We have clearly described our model-building procedure in our method section.**

**Thanks again for your excellent instruction.**